# Analysis of the different interventions scenario for programmatic measles control in Bangladesh: A modelling study

**Md Abdul Kuddus**[1,2]*, **Azizur Rahman**[3], **Farzana Alam**[4], **M. Mohiuddin**[5]

**1** Australian Institute of Tropical Health and Medicine, James Cook University, Townsville, QLD, Australia, **2** Department of Mathematics, University of Rajshahi, Rajshahi, Bangladesh, **3** School of Computing and Mathematics, Charles Sturt University, Wagga Wagga, NSW, Australia, **4** Department of Electrical and Electronic Engineering, Rajshahi University of Engineering & Technology, Rajshahi, Bangladesh, **5** Department of Mathematics, Comilla University, Cumilla, Bangladesh

* mdabdul.kuddus@my.jcu.edu.au

**Data Availability Statement:** All data are fully available within the paper and Supporting Information files.

## Abstract

In recent years measles has been one of the most critical public health problem in Bangladesh. Although the Ministry of Health in Bangladesh employs a broad extension of measles control policies, logistical challenges exist, and there is significant doubt regarding the disease burden. Mathematical modelling of measles is considered one of the most effective ways to understand infection transmission and estimate parameters in different countries, such as Bangladesh. In this study, a mathematical modelling framework is presented to explore the dynamics of measles in Bangladesh. We calibrated the model using cumulative measles incidence data from 2000 to 2019. Also, we performed a sensitivity analysis of the model parameters and found that the contact rate had the most significant influence on the basic reproduction number $R_0$. Four hypothetical intervention scenarios were developed and simulated for the period from 2020 to 2035. The results show that the scenario which combines enhanced treatment for exposed and infected population, first and second doses of vaccine is the most effective at rapidly reducing the total number of measles incidence and mortality in Bangladesh. Our findings also suggest that strategies that focus on a single interventions do not dramatically affect the decline in measles incidence cases; instead, those that combine two or more interventions simultaneously are the most effective in decreasing the burden of measles incidence and mortality. In addition, we also evaluated the cost-effectiveness of varying combinations of three basic control strategies including distancing, vaccination and treatment, all within the optimal control framework. Our finding suggested that combines distancing, vaccination and treatment control strategy is the most cost-effective for reducing the burden of measles in Bangladesh. Other strategies can be comprised to measles depending on the availability of funds and policymakers' choices.

## 1. Introduction

Some death-dealing contagious diseases infected many people in different areas of the world in the last few decades. Measles, a human respiratory disease, is one of these devastating

**Funding:** This work was not funded and did not receive any specific grant from funding agencies in the public, commercial, or not-for profit sectors.

**Competing interests:** The authors declare that they have no known competing financial interests or personal relationships that could have appeared to influence the work reported in this paper.

human infectious diseases caused by a paramoxyviridae family virus [1]. The measles virus is naturally found in the human body, and it may be life-threatening for young children under the age of five years and adults older than the age of twenty years [2]. For developing the initial symptoms in the body of infected persons, measles takes 10–14 days as incubation time after the exposure of the virus, and the infected persons usually get well from illness in three weeks without any complications [3]. However, people who have a deficiency of vitamin A or suffer from malnutrition may face complications like diarrhea, ear infection, brain inflammation, pneumonia and blindness [4]. The clinical symptoms of measles are high fever, cough, runny nose, red eyes, sore throat, white spot inside the mouth, and a general skin rush initially appearing on the head and slowly spreading to other parts of the body [5]. Measles is a highly devastating infectious virus that means the infection can transmit directly from person to person through the coughing and sneezing of infected persons. It can even remain infectious in the air or on the surface for up to two hours, and if other people breathe this contaminated air or touch the surface, they must be infected [6].

There is no fixed medicine to recover the measles infected person. Patients with measles are suggested for bed rest, fluids, control of fever, and antibiotics [7]. The measles vaccine was first introduced in 1963. Since then, vaccination is the only fruitful way to control the measles epidemic, and about 73% of measles-related deaths were declined globally from 2000 to 2018 due to the vaccination [3]. According to the World Health Organization (WHO) recommendation, about 85% immunity is produced after the first dose of vaccine at the age of 9–11 months child. This percentage of immunity increases to 97% after the second dose of vaccine at greater than 12 months [8]. Despite the availability of this safe, effective and inexpensive vaccine, measles outbreaks have occurred in different parts of the world several times. In the USA, California faced a very worst situation of measles outbreak from 1988 to 1990 with 16,400 reported cases and 75 deaths [9]. In 2019, more than 1249 measles cases were reported in the USA, which was the largest number of cases since 1992 [10]. In the African region, Congo experienced its worst measles outbreak in 2019, with 250270 reported cases and 5110 deaths [11]. Some European and Eastern Mediterranean countries like Ukraine, Kazakhastan, Georgia, Yemen, Somalia and Sudan also faced a large measles outbreak in 2019 [11]. In 2018, a total of 50,000 people were infected, and about 300 people died in Madagascar; most of them were children under five years of age [12]. Globally, WHO estimated 97,69,400 confirmed measles cases with more than 1,40,000 deaths in 2018 and a total of 8,69,770 cases with 2,07,500 deaths in 2019 [13].

However, despite being vaccine-preventable, measles is still an issue of concern of public health department in many developing Asian and African countries due to low awareness, civil strife, vaccine hesitancy, and lower immunization system. Bangladesh, an overpopulated and poor economic Asian country, has experienced measles outbreaks in different areas between 2000 and 2019. The World Health Organization (WHO) estimated about 14,877 measles cases in 2005, 5,329 cases in 2011, and 1,793 cases in 2012 in Bangladesh [14]. To control and prevent the measles outbreak, the Bangladesh government took some initiatives like nationwide vaccination, additional supplementary immunization activities, and strengthening the case-based surveillance system in 2005, 2006, 2010 and 2014 [14]. The measles cases declined at 6 in 2005 and 250 in 2015 for the government adopted activities. However, it started to increase again from 2016. According to the estimation of WHO, in 2019, there were 4,181 measles cases in Bangladesh, with most of them were from the Rohingya refugee camp in Cox's Bazaar [11]. A total of 1,724 measles cases were reported in 2017 and approximately 1,319 cases in 2018 [15]. Due to these increasing scenarios of measles cases, Bangladesh has still considered the measles affected country.

The concept of mathematical modeling is used to describe different real-world phenomena's. The researchers have been using mathematical modeling in epidemiology since the middle of the 20th century to interpret the mechanism of spread and find ways of controlling and eliminating infectious diseases. Some renowned researchers have already done a few important studies on measles transmission dynamics deterministically using various essential compartments in recent decades. In Ochoche and Gweryina [16], a SIR deterministic compartmental model of measles was studied, highlighting the impact of vaccination. The authors realized that it is possible to eliminate measles through vaccination, and they advised that at least two doses of vaccine should be mandatory for all children. For controlling the transmission dynamics of measles, an optimal vaccine coverage level was investigated by Momoh *et al.* [17]. Edward *et al.* discussed a deterministic mathematical model to eradicate and control the measles pandemic [6]. They observed that the measles would be died out if the basic reproduction number is less than one, i.e. $R_0 < 1$. In another study Fred, Sigey [18], the authors performed on an SEIR mathematical model with vaccination, and they suggested that the mass vaccination and the detection of measles cases as early as possible should be introduced to abet transmission.

Additionally, Christopher *et al.* studied a mathematical measles model with vaccination and measles drug therapy [19]. They found that the vaccination in susceptible class and drug therapy in exposed class to identified infected person eliminated measles more efficiently. In the study [20], another SEIR mathematical model also investigated to control the transmission dynamics of measles. Although they discussed stability, disease-free and endemic equilibriums, they did not address the impact of the vaccination in their study. In the abovementioned studies, all researchers have developed a deterministic mathematical measles model considering different compartments, including vaccination, to describe the transmission dynamics of measles. Nevertheless, no research has not been done so far focusing on the mathematical model of the mechanism of the spread of measles with double dose vaccination and different intervention scenarios for programmatic measles control.

In this study, we developed a measles model with a double dose of vaccination to describe the transmission dynamics of measles in Bangladesh. The model is calibrated to the Bangladesh demographic and measles incidence data from 2000 to 2019 to estimate the key transmission and progression parameters. Multiple intervention scenarios were considered to explore the impact of each on its own and when combined on measles incidence and mortality. We also performed cost-effective analysis to explore which intervention will be most cost-effective compare to others. This study depicts Bangladesh specific elimination strategies and describes the results of different levels of investment in future on measles control.

The entire paper is designed as follows: Section 2 introduces model development, basic reproduction number estimation, parameter estimation and sensitivity analysis. In Sections 3 and 4, we discussed the development of different scenarios and cost-effective strategy. Finally, section 5 contains discussion and conclusions.

## 2. Methods and materials

### 2.1 Model development

We developed a compartmental transmission dynamics measles model between the following mutually exclusive compartments: susceptible individuals, S(t); those who have not yet infected with the disease but might become infected; first dose vaccinated individuals, $V_1(t)$; those who have received the first dose of vaccine; second dose vaccinated individuals, $V_2(t)$; those who have received the second dose of vaccine; Exposed individuals, E(t); representing those that are infected and have not yet developed active measles disease; Infected individuals,

I(t); those who are infected and infectious; and Recovered individuals, R(t); those who were previously infected and successfully recovered.

The total population size N(t) is assumed to be constant and well mixed:

$$N(t) = S(t) + V_1(t) + V_2(t) + E(t) + I(t) + R(t) \tag{1}$$

To ensure the population size constant, we replace all deaths as newborns in the susceptible compartment. It includes death through natural causes, which occurs in all states at the constant per-capita rate μ, and measles-related deaths, which occur at the constant per-capita rate δ. Susceptible population (S) who receive the first dose of vaccine move to the vaccinated compartment at a rate η. The first dose of vaccinated population $V_1$ moves to the susceptible compartment at a rate ρ, and the rest of the population moves to the second dose of vaccinated population $V_2$ at a per-capita rate σ. The second dose of the vaccinated population also moves to the recovery compartment at a rate ω. Individuals in the S compartment may be infected with the measles virus at a rate λ = βSI, where β is the transmission rate between infected and susceptible population. Once infected, individuals move to the exposed compartment E. A proportion of the exposed population progress to the infected compartment at a per-capita rate α, and the rest of the exposed population progress to the recovered compartment at a per-capita rate κ due to the treatment of the exposed population. The proportion of the infected individuals move to the recovery compartment due to the treatment rate τ and natural recovery rate γ. The model flow diagram is presented in Fig 1.

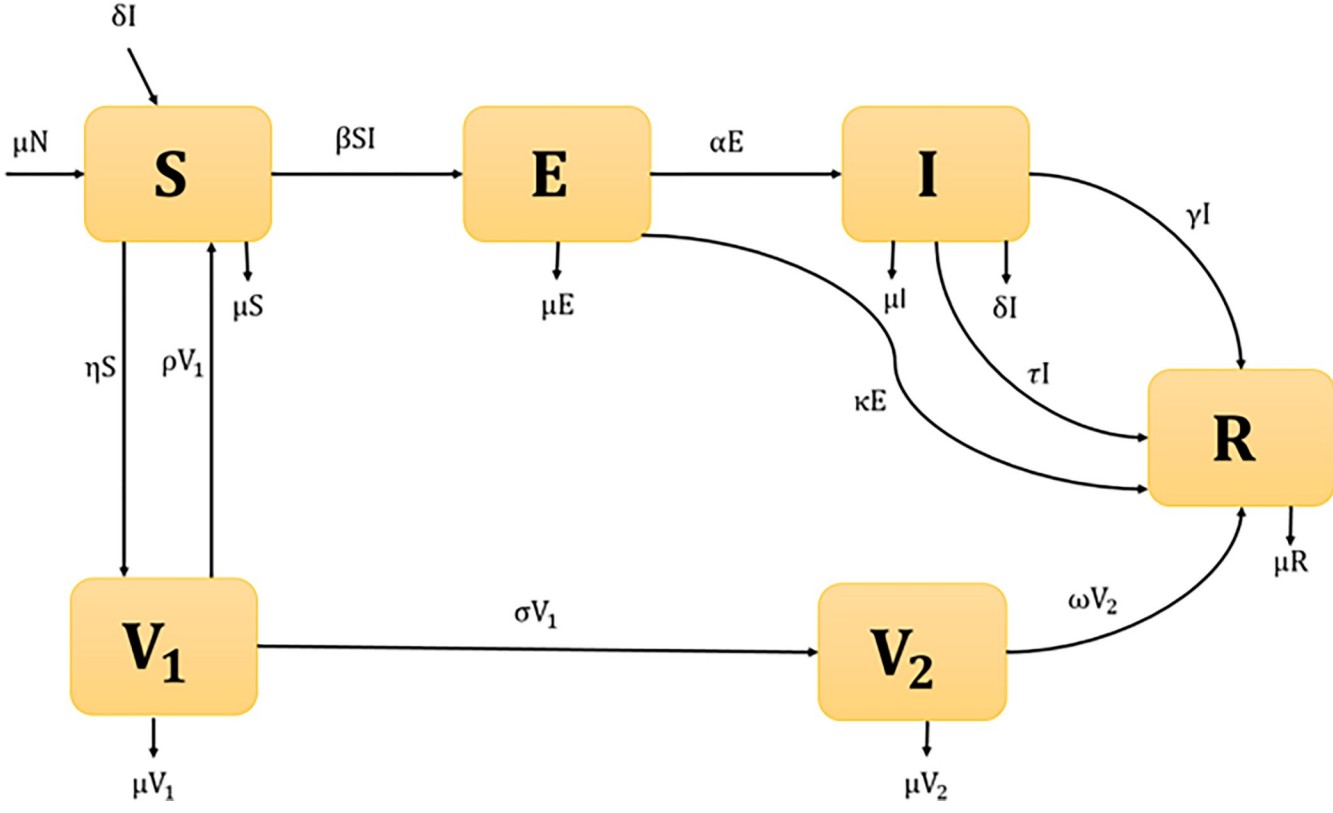

**Fig 1. Compartmental model of measles transmission in Bangladesh setting.**

From the above-mentioned, the transmission dynamics of measles is given by the following compartmental system of nonlinear ordinary differential equations that describe the model:

$$\frac{dS}{dt} = \mu N - \beta SI - \eta S - \mu S + \delta I + \rho V_1 \tag{2}$$

$$\frac{dV_1}{dt} = \eta S - \rho V_1 - \sigma V_1 - \mu V_1 \tag{3}$$

$$\frac{dV_2}{dt} = \sigma V_1 - \omega V_2 - \mu V_2 \tag{4}$$

$$\frac{dE}{dt} = \beta SI - \alpha E - \kappa E - \mu E \tag{5}$$

$$\frac{dI}{dt} = \alpha E - \gamma I - \delta I - \tau I - \mu I \tag{6}$$

$$\frac{dR}{dt} = \gamma I + \omega V_2 + \kappa E + \tau I - \mu R \tag{7}$$

The initial conditions of the system (2)–(7) are of the form

$$S(0) \geq 0, V_1(0) \geq 0, V_2(0) \geq 0, E(0) \geq 0, I(0) \geq 0, R(0) \geq 0. \tag{8}$$

It can be easily shown that the solution of the system (2)–(7) subject to the initial conditions (8) exists and is nonnegative for all t≥0.

## 2.2 Basic reproduction number

The basic reproduction number can be determined using the method of next-generation matrix [21]. The next-generation matrix is the production of matrices T and $-\Sigma^{-1}$ where the matrix T represents the rate of infection transmission in E and I compartments and the matrix $\Sigma$ describes all other transfer across the compartments. The matrices T and $\Sigma$ are given as

$$T = \begin{pmatrix} 0 & \beta S^0 \\ 0 & 0 \end{pmatrix} \text{ and } \Sigma = \begin{pmatrix} -(\alpha + \kappa + \mu) & 0 \\ \alpha & -(\gamma + \delta + \tau + \mu) \end{pmatrix}$$

The next-generation matrix is

$$K = T \times \left(-\Sigma^{-1}\right) = \begin{pmatrix} 0 & \beta S^0 \\ 0 & 0 \end{pmatrix} \times \begin{pmatrix} \dfrac{1}{(\alpha + \kappa + \mu)} & 0 \\ \dfrac{\alpha}{(\alpha + \kappa + \mu)(\gamma + \delta + \tau + \mu)} & \dfrac{1}{(\gamma + \delta + \tau + \mu)} \end{pmatrix}$$

$$= \begin{pmatrix} \dfrac{\beta S^0 \alpha}{(\alpha + \kappa + \mu)(\gamma + \delta + \tau + \mu)} & \dfrac{\beta S^0}{(\gamma + \delta + \tau + \mu)} \\ 0 & 0 \end{pmatrix}$$

The basic reproduction number is the Eigen-value of largest magnitude of the next-generation matrix (K). Hence the basic reproduction number is obtained as

$$R_0 = \frac{\beta S^0 \alpha}{(\alpha + \kappa + \mu)(\gamma + \delta + \tau + \mu)} = \frac{\alpha \beta \mu N(\rho + \sigma + \mu)}{(\alpha + \kappa + \mu)(\gamma + \delta + \tau + \mu)((\eta + \mu)(\rho + \sigma + \mu) - \rho \eta)}$$

## 2.3 Parameter estimation

We estimated the measles model parameters from fitting different combinations of parameters in Eqs (2)–(7) to the actual number of measles cases in Bangladesh from 2000 to 2019 [22]. In order to parameterize measles model (2)–(7), we obtained some of the parameter values from the literature (see Table 1), rest of the parameters were estimated from data fitting (see Fig 2). The estimation of parameters was carried out using the least-squares method, which minimises summation of the square errors given by $\Sigma(A(t,y) - B_{actual})^2$ subject to the measles model (2)–(7), where $B_{actual}$ is the actual reported measles data, and $A(t,y)$ denotes the solution of the model corresponding to the number of measles cases over time t with the set of estimated parameters, denoted by y.

## 2.4 Sensitivity analysis

We perform the sensitivity of the model basic reproduction number ($R_0$) to the model parameters using the Latin Hypercube Sampling (LHS) method with 10000 runs per simulation. The LHS is a Monte Carlo stratified sampling technique that permits us to concurrently achieve an unbiased assessment of the model output for a particular set of input parameter values. The Partial Rank Correlation Coefficients (PRCCs) for the full range of parameters are shown in Fig 3. The PRCCs for the basic reproduction number in Fig 3 is produced using the expressions $R_0$. Results show that parameters transmission rate (β) and progression rate (α) have a positive correlation with the model outcomes $R_0$, which means that decreasing these parameters values will reduce the prevalence of measles. On the other hand, parameters ρ, σ, γ, κ, τ and η have a negative correlation with the model outcomes $R_0$, which indicates that increasing these parameters will decrease the outbreak of measles.

## 2.5 Ethical approval

This study based on aggregated measles surveillance data in Bangladesh taken from the World Health Organization. No confidential information included because analyses were performed at the aggregate level. Therefore, no ethical approval is required.

**Table 1. Depiction and estimation of the measles model (2)–(7) parameters.**

| Parameters | Description | Estimated value | References |
|---|---|---|---|
| N | Total population in Bangladesh | 163,046,161 | [23] |
| μ | Per-capita death rate | $\frac{1}{70}$ per year | [24] |
| β | Transmission rate | $6.99 \times 10^{-7}$ | Fitted |
| η | First dose of vaccination rate | 0.94 | [25] |
| ρ | Progression rate from S to $V_1$ | 0.6 | [2] |
| σ | Second dose of vaccination rate | 0.93 | [25] |
| ω | Recovery rate due to the second dose of vaccine | 0.8 | [2] |
| α | Progression rate from E to I | 0.019 | Fitted |
| δ | Measles related death rate | 0.125 | [2] |
| γ | Natural recovery rate | 0.6 | [2] |
| κ | Treatment rate for exposed population | 0.08 | [26] |
| τ | Treatment rate for infected population | 0.14 | [26] |

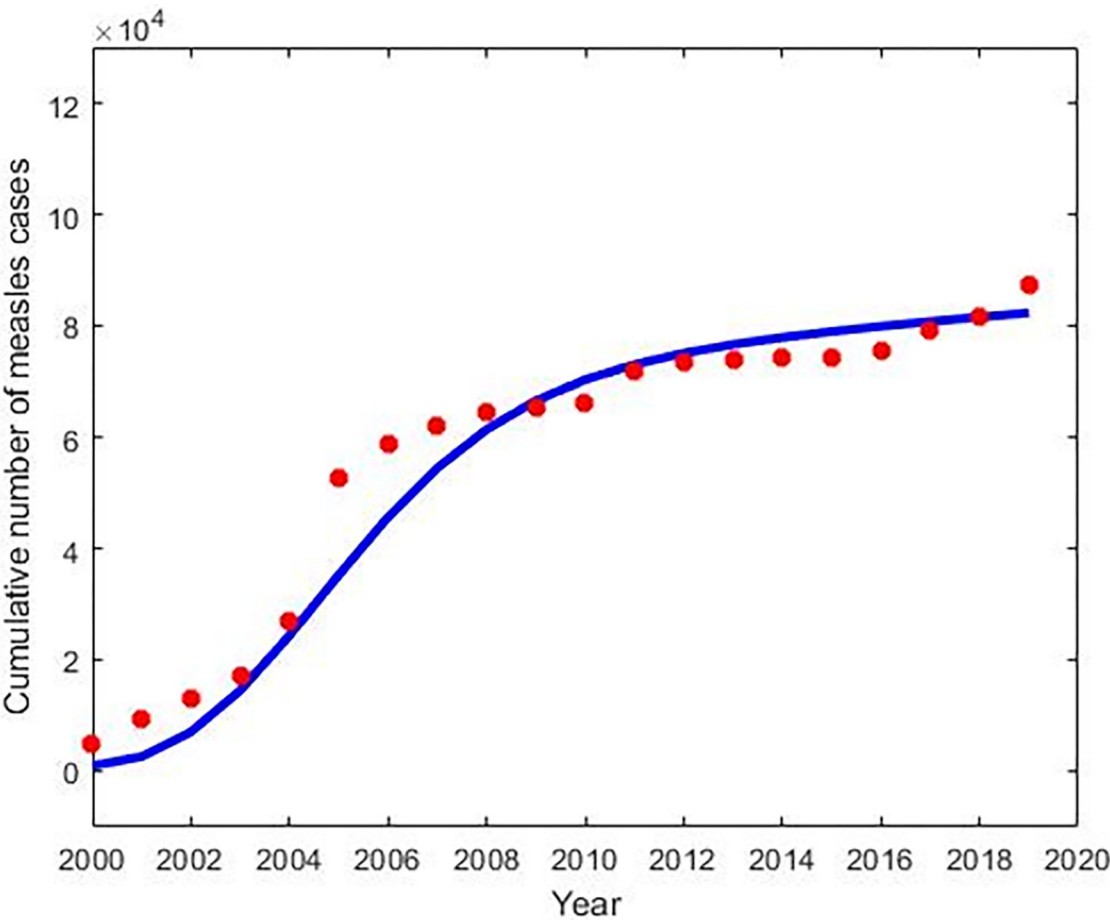

**Fig 2. Cumulative number of confirmed measles cases from 2000 to 2019 (red dot) and the corresponding model best fit (blue solid curve) in Bangladesh.**

## 3. Scenario development and analysis

This section developed four potential intervention scenarios to explore the dynamics of measles incidence and mortality in Bangladesh. These scenarios are detailed in Table 2, 3, 4 and 5. We parameterized these proposed responses to our model structure to assess the effect of these responses during the period 2020–2035. In 2014, Bangladesh, one of 11 countries in the South-East Asia Region, adopted a national goal for measles elimination by 2018. In Bangladesh the Ministry of Health achieved its objectives in first and second doses vaccines, effective treatment and overall management through partnership, engaging all care providers (GO-NGOs) and making available free vaccination support. Estimated coverage with the first dose vaccine increased from 74% in 2000 to 94% in 2016. The second dose vaccine was introduced in 2012, and its coverage increased from 35% in 2013 to 93% in 2016 [25]. In Bangladesh, treatment coverage for asymptomatic and symptomatic measles patients are around 8% and 14%, respectively [26]. Therefore, it is essential to increases the treatment rate for both asymptomatic and symptomatic measles patients in Bangladesh.

Scenario 1 simulates a continuation of the programmatic situation during the period 2020–2035. During this time, we incorporated four different intervention strategies: increasing both first and second dose vaccine; improving treatment rates for exposed (asymptomatic) and infected (symptomatic) population. We implement these as single interventions and compare

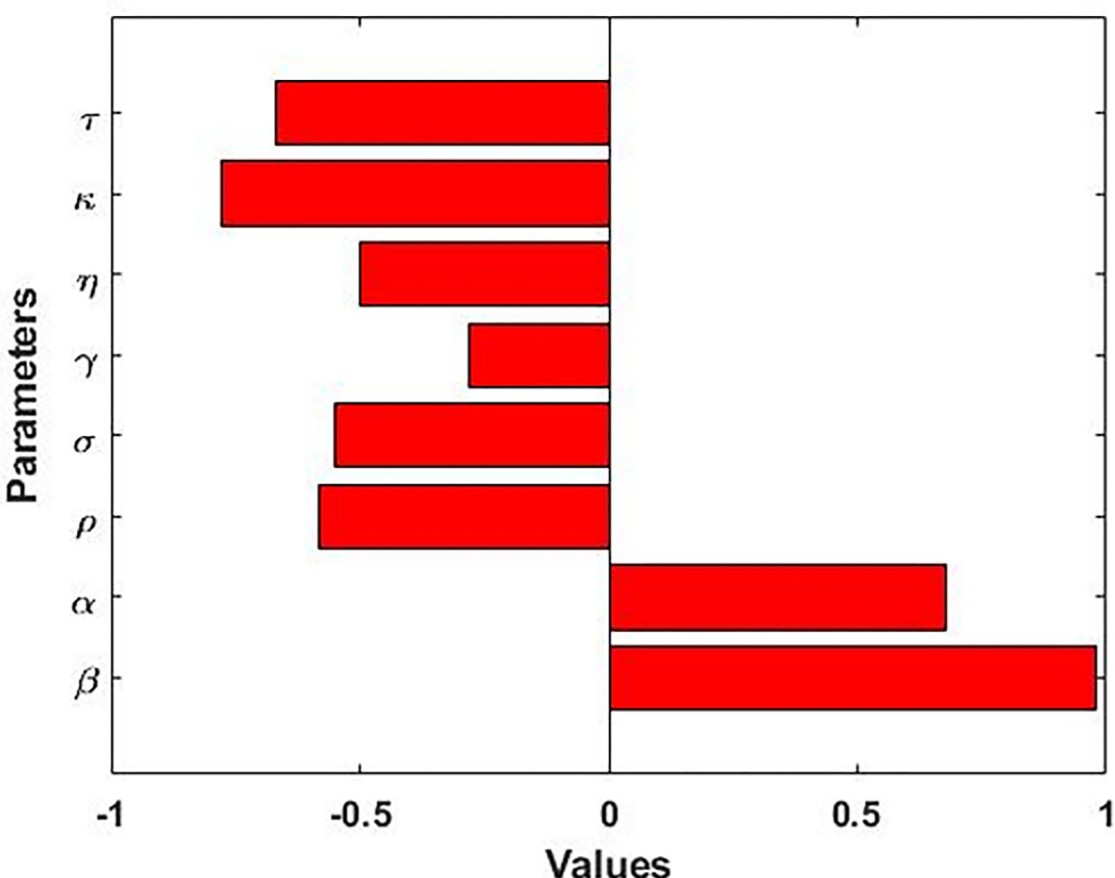

**Fig 3. PRCC values depicting the sensitivity of the basic reproduction number $R_0$ with respect to the parameters β, α, ρ, σ, γ, η, κ and τ.**

them with baseline (see Table 2) to explore the impact of each intervention on total measles incidence and mortality. In this scenario, the first and second dose vaccine increase from 94% and 93% to 100% and 100%, respectively. Further, treatment rates for asymptomatic and symptomatic cases improved through a combination of case-finding strategies and improved knowledge of standard operating procedures for measles treatment commencement. The case finding is to identify of symptomatic patients attending a health facility, either their initiative or referred by another health facility, health worker, and community volunteer. These strategies improve the treatment rate for an asymptomatic case from around 8% to 100% and symptomatic cases from around 14% to 100%.

Scenario 2 consists of a combination of six-double intervention strategies, which include (i) enhanced first dose and second dose vaccine, (ii) enhanced the first dose vaccine and treatment rate for exposed population, (iii) enhanced first dose vaccine and treatment rate for infected population, (iv) improved second dose vaccine and treatment rate for an exposed population, (v) improved second dose vaccine and treatment rate for an infected population, and (vi) enhanced treatment rate for exposed and infected population.

Scenario 3 incorporates a combination of four-triple intervention strategies such as (i) first and second dose vaccine as well as treatment rate for an exposed population, (ii) first and second dose vaccine as well as treatment rate for an infected population, (iii) first dose vaccine, treatment rate for an exposed and infected population, (iv) second dose vaccine, treatment

**Table 2. Hypothetical single intervention strategy implemented in our proposed model of measles control in Bangladesh, for the period 2020–2035.**

| Parameters | Parameter values | Estimated measles actual incident cases | Reduction from baseline | Estimated measles actual mortality | Reduction from baseline |
|---|---|---|---|---|---|
| First dose vaccine rate (η) | Baseline (94%) | $3.714 \times 10^5$ | $0.000 \times 10^5$ | $5.546 \times 10^4$ | $0.000 \times 10^4$ |
| | 96% | $3.686 \times 10^5$ | $0.028 \times 10^5$ | $5.507 \times 10^4$ | $0.039 \times 10^4$ |
| | 98% | $3.686 \times 10^5$ | $0.028 \times 10^5$ | $5.507 \times 10^4$ | $0.039 \times 10^4$ |
| | 99% | $3.686 \times 10^5$ | $0.028 \times 10^5$ | $5.507 \times 10^4$ | $0.039 \times 10^4$ |
| | 100% | $3.686 \times 10^5$ | $0.028 \times 10^5$ | $5.507 \times 10^4$ | $0.039 \times 10^4$ |
| Second dose vaccine rate (σ) | Baseline (93%) | $3.714 \times 10^5$ | $0.000 \times 10^5$ | $5.546 \times 10^4$ | $0.000 \times 10^4$ |
| | 95% | $3.714 \times 10^5$ | $0.000 \times 10^5$ | $5.546 \times 10^4$ | $0.000 \times 10^4$ |
| | 97% | $3.714 \times 10^5$ | $0.000 \times 10^5$ | $5.546 \times 10^4$ | $0.000 \times 10^4$ |
| | 99% | $3.714 \times 10^5$ | $0.000 \times 10^5$ | $5.546 \times 10^4$ | $0.000 \times 10^4$ |
| | 100% | $3.714 \times 10^5$ | $0.000 \times 10^5$ | $5.546 \times 10^4$ | $0.000 \times 10^4$ |
| Treatment rate for exposed population (κ) | Baseline (8%) | $3.714 \times 10^5$ | $0.000 \times 10^5$ | $5.546 \times 10^4$ | $0.000 \times 10^4$ |
| | 25% | $0.524 \times 10^5$ | $3.190 \times 10^5$ | $0.903 \times 10^4$ | $4.643 \times 10^4$ |
| | 50% | $0.031 \times 10^5$ | $3.683 \times 10^5$ | $0.677 \times 10^3$ | $5.478 \times 10^4$ |
| | 75% | $0.002 \times 10^5$ | $3.712 \times 10^5$ | $0.732 \times 10^2$ | $5.537 \times 10^4$ |
| | 100% | $0.003 \times 10^4$ | $3.7137 \times 10^5$ | $0.138 \times 10^2$ | $5.545 \times 10^4$ |
| Treatment rate for infected population (τ) | Baseline (14%) | $3.714 \times 10^5$ | $0.000 \times 10^5$ | $5.546 \times 10^4$ | $0.000 \times 10^4$ |
| | 25% | $3.512 \times 10^5$ | $0.202 \times 10^5$ | $4.634 \times 10^4$ | $0.912 \times 10^4$ |
| | 50% | $3.150 \times 10^5$ | $0.564 \times 10^5$ | $3.306 \times 10^4$ | $2.240 \times 10^4$ |
| | 75% | $2.873 \times 10^5$ | $0.841 \times 10^5$ | $2.501 \times 10^4$ | $3.045 \times 10^4$ |
| | 100% | $2.656 \times 10^5$ | $1.058 \times 10^5$ | $1.977 \times 10^4$ | $3.569 \times 10^4$ |

rate for exposed and infected population. Finally, scenario 4 describes the broad scale-up combination of first and second dose vaccines and the treatment rate for exposed and infected populations. Under this scenario, all model parameters values changed simultaneously and considered overall scale-up values for each parameter. However, each category of intervention could involve several potential specific activities. For example, treatment for exposed and infected populations could include training of doctors, nurses, and pharmacists on measles guidelines, monitoring and managing supplies of high-quality drugs. We assumed that these different activities would have a similar impact on the model and did not model the impact of these explicit activities independently.

The estimated outcomes in this study, over a 15-year time frame, included: first dose and second dose vaccine, treatment for exposed and infected population. To better understand the contribution of changing specific interventions with different scenarios, estimated outcomes were presented separately for each intervention assumed to be influenced by the scenario. Results from scenario one are presented in Figs 4 and 5 as well as Table 2. From scenario 1, we observed that within a four-single intervention strategy, the treatment rate for the exposed population is the most effective than other single interventions, which reduces more measles incidence and mortality (see Figs 4(A3) and 5(B3), and Table 2) in Bangladesh. Hence, it is the preferable single intervention strategy. Alternatively, the treatment rate for the infected population is another option.

Figs 6 and 7, Table 3, represent scenario 2, which includes our proposed double interventions strategies. Each of the interventions resulted in decreasing the number of measles incidence and mortality. The analysis shows that a combination treatment rates of the exposed and infected population are the best dual intervention strategy for reducing the number of

**Table 3. Hypothetical double intervention strategy implemented in our proposed model of measles control in Bangladesh, for the period 2020–2035.**

| Parameters | Parameter values | Estimated measles actual incident cases | Reduction from baseline | Estimated measles actual mortality | Reduction from baseline |
|---|---|---|---|---|---|
| First dose ($\eta$) and second dose ($\sigma$) vaccine rates | Baseline (94% & 93%) | $3.714 \times 10^5$ | $0.000 \times 10^5$ | $5.546 \times 10^4$ | $0.000 \times 10^4$ |
| | 96% and 95% | $3.676 \times 10^5$ | $0.038 \times 10^5$ | $5.494 \times 10^4$ | $0.052 \times 10^4$ |
| | 98% and 97% | $3.666 \times 10^5$ | $0.048 \times 10^5$ | $5.481 \times 10^4$ | $0.065 \times 10^4$ |
| | 99% and 99% | $3.656 \times 10^5$ | $0.058 \times 10^5$ | $5.468 \times 10^4$ | $0.078 \times 10^4$ |
| | 100% and 100% | $3.652 \times 10^5$ | $0.062 \times 10^5$ | $5.462 \times 10^4$ | $0.084 \times 10^4$ |
| First dose ($\eta$) and treatment rate for exposed population ($\kappa$) | Baseline (94% & 8%) | $3.714 \times 10^5$ | $0.000 \times 10^5$ | $5.546 \times 10^4$ | $0.000 \times 10^4$ |
| | 96% & 25% | $0.518 \times 10^5$ | $3.196 \times 10^5$ | $0.894 \times 10^4$ | $4.652 \times 10^4$ |
| | 98% and 50% | $0.030 \times 10^5$ | $3.684 \times 10^5$ | $0.694 \times 10^3$ | $5.476 \times 10^4$ |
| | 99% and 75% | $0.246 \times 10^3$ | $3.712 \times 10^5$ | $0.734 \times 10^2$ | $5.539 \times 10^4$ |
| | 100% and 100% | $0.348 \times 10^2$ | $3.714 \times 10^5$ | $0.139 \times 10^2$ | $5.545 \times 10^4$ |
| First dose ($\eta$) and treatment rate for infected population ($\tau$) | Baseline (94% & 14%) | $3.714 \times 10^5$ | $0.000 \times 10^5$ | $5.546 \times 10^4$ | $0.000 \times 10^4$ |
| | 96% and 25% | $3.484 \times 10^5$ | $0.230 \times 10^5$ | $4.599 \times 10^4$ | $0.947 \times 10^4$ |
| | 98% and 50% | $3.122 \times 10^5$ | $0.592 \times 10^5$ | $3.279 \times 10^4$ | $2.267 \times 10^4$ |
| | 99% and 75% | $2.848 \times 10^5$ | $0.866 \times 10^5$ | $2.483 \times 10^4$ | $3.063 \times 10^4$ |
| | 100% and 100% | $2.635 \times 10^5$ | $1.079 \times 10^5$ | $1.960 \times 10^4$ | $3.586 \times 10^4$ |
| Second dose ($\sigma$) and treatment rate for exposed population ($\kappa$) | Baseline (93% and 8%) | $3.714 \times 10^5$ | $0.000 \times 10^5$ | $5.546 \times 10^4$ | $0.000 \times 10^4$ |
| | 95% and 25% | $0.524 \times 10^5$ | $3.190 \times 10^5$ | $0.903 \times 10^4$ | $4.643 \times 10^4$ |
| | 97% and 50% | $0.031 \times 10^5$ | $3.683 \times 10^5$ | $0.677 \times 10^3$ | $5.478 \times 10^4$ |
| | 99% and 75% | $0.244 \times 10^3$ | $3.712 \times 10^5$ | $0.732 \times 10^2$ | $5.539 \times 10^4$ |
| | 100% and 100% | $0.343 \times 10^2$ | $3.714 \times 10^5$ | $0.137 \times 10^2$ | $5.545 \times 10^4$ |
| Second dose ($\sigma$) and treatment rate for infected population ($\tau$) | Baseline (93% and 14%) | $3.714 \times 10^5$ | $0.000 \times 10^5$ | $5.546 \times 10^4$ | $0.000 \times 10^4$ |
| | 95% and 25% | $3.502 \times 10^5$ | $0.212 \times 10^5$ | $4.622 \times 10^4$ | $1.924 \times 10^4$ |
| | 97% and 50% | $3.131 \times 10^5$ | $0.583 \times 10^5$ | $3.288 \times 10^4$ | $2.258 \times 10^4$ |
| | 99% and 75% | $2.848 \times 10^5$ | $0.866 \times 10^5$ | $2.479 \times 10^4$ | $3.067 \times 10^4$ |
| | 100% and 100% | $2.628 \times 10^5$ | $1.086 \times 10^5$ | $1.957 \times 10^4$ | $3.589 \times 10^4$ |
| Treatment rate for exposed population ($\kappa$) and treatment rate for infected population ($\tau$) | Baseline (8% and 14%) | $3.714 \times 10^5$ | $0.000 \times 10^5$ | $5.546 \times 10^4$ | $0.000 \times 10^4$ |
| | 25% and 25% | $0.455 \times 10^5$ | $3.259 \times 10^5$ | $0.689 \times 10^4$ | $4.857 \times 10^4$ |
| | 50% and 50% | $0.013 \times 10^5$ | $3.701 \times 10^5$ | $0.188 \times 10^3$ | $5.527 \times 10^4$ |
| | 75% and 75% | $0.311 \times 10^2$ | $3.713 \times 10^5$ | $0.417 \times 10^1$ | $5.5455 \times 10^4$ |
| | 100% and 100% | $0.070 \times 10^1$ | $3.714 \times 10^5$ | $0.098 \times 10^0$ | $5.5459 \times 10^4$ |

measles incidence and mortality in Bangladesh (see Figs 6(A6) and 7(B6), and Table 3). Alternative, the combination of first dose vaccination rate and treatment rate for an exposed population is another option.

Figs 8 and 9 and Table 4 depict scenario 3, which include a combination of triple interventions strategies. As expected, each of the strategies decreased the number of measles incidence and mortality in Bangladesh. The combination of first dose vaccination rate, treatment rates for the exposed and infected population is the best triple intervention strategy (see Figs 8(A3) and 9(B3), and Table 4). Alternative, second dose vaccination rate, and treatment rates for exposed and infected populations are also better triple intervention strategies.

**Table 4. Hypothetical triple intervention strategy implemented in our proposed model of measles control in Bangladesh, for the period 2020–2035.**

| Parameters | Parameter values | Estimated measles actual incident cases | Reduction from baseline | Estimated measles actual mortality | Reduction from baseline |
|---|---|---|---|---|---|
| $\eta$, $\sigma$ and $\kappa$ | Baseline (94%, 93% & 8%) | $3.714 \times 10^5$ | $0.000 \times 10^5$ | $5.546 \times 10^4$ | $0.000 \times 10^4$ |
| | 96%, 95% & 25% | $0.515 \times 10^5$ | $3.199 \times 10^5$ | $0.889 \times 10^4$ | $4.657 \times 10^4$ |
| | 98%, 97% & 50% | $0.029 \times 10^5$ | $3.685 \times 10^5$ | $0.661 \times 10^3$ | $5.479 \times 10^4$ |
| | 99%, 99% & 75% | $0.235 \times 10^3$ | $3.712 \times 10^5$ | $0.709 \times 10^2$ | $5.539 \times 10^4$ |
| | 100%, 100% & 100% | $0.329 \times 10^2$ | $3.714 \times 10^5$ | $0.133 \times 10^2$ | $5.535 \times 10^4$ |
| $\eta$, $\sigma$ and $\tau$ | Baseline (94%, 93% & 14%) | $3.714 \times 10^5$ | $0.000 \times 10^5$ | $5.546 \times 10^4$ | $0.000 \times 10^4$ |
| | 96%, 95% & 25% | $3.474 \times 10^5$ | $0.240 \times 10^5$ | $4.587 \times 10^4$ | $0.959 \times 10^4$ |
| | 98%, 97% & 50% | $3.104 \times 10^5$ | $0.610 \times 10^5$ | $3.261 \times 10^4$ | $2.285 \times 10^4$ |
| | 99%, 99% & 75% | $2.823 \times 10^5$ | $0.891 \times 10^5$ | $2.459 \times 10^4$ | $3.087 \times 10^4$ |
| | 100%, 100% & 100% | $2.605 \times 10^5$ | $1.109 \times 10^5$ | $1.941 \times 10^4$ | $3.605 \times 10^4$ |
| $\eta$, $\tau$ and $\kappa$ | Baseline (94%, 14% & 8%) | $3.714 \times 10^5$ | $0.000 \times 10^5$ | $5.546 \times 10^4$ | $0.000 \times 10^4$ |
| | 96%, 25% & 25% | $0.447 \times 10^5$ | $3.267 \times 10^5$ | $0.679 \times 10^4$ | $4.867 \times 10^4$ |
| | 98%, 50% & 50% | $0.013 \times 10^5$ | $3.701 \times 10^5$ | $0.183 \times 10^3$ | $5.527 \times 10^4$ |
| | 99%, 75% & 75% | $0.305 \times 10^2$ | $3.7136 \times 10^5$ | $0.404 \times 10^1$ | $5.5455 \times 10^4$ |
| | 100%, 100% & 100% | $0.068 \times 10^1$ | $3.7139 \times 10^5$ | $0.095 \times 10^0$ | $5.5459 \times 10^4$ |
| $\sigma$, $\tau$ and $\kappa$ | Baseline (93%, 14% & 8%) | $3.714 \times 10^5$ | $0.000 \times 10^5$ | $5.546 \times 10^4$ | $0.000 \times 10^4$ |
| | 95%, 25% & 25% | $0.452 \times 10^5$ | $3.262 \times 10^5$ | $0.686 \times 10^4$ | $4.860 \times 10^4$ |
| | 97%, 50% & 50% | $0.013 \times 10^5$ | $3.701 \times 10^5$ | $0.186 \times 10^3$ | $5.527 \times 10^4$ |
| | 99%, 75% & 75% | $0.303 \times 10^2$ | $3.7136 \times 10^5$ | $0.406 \times 10^1$ | $5.5455 \times 10^4$ |
| | 100%, 100% & 100% | $0.068 \times 10^1$ | $3.7139 \times 10^5$ | $0.095 \times 10^0$ | $5.5459 \times 10^4$ |

Finally, Fig 10 and Table 5 represents scenario 4, which includes first and second dose vaccination rates and treatment rates for exposed and infected populations. Under scenario 4, measles incidence and mortality reduce enormously over the 15-years due to the combination of immensely expanding first and second dose vaccination rates and treatment rates for exposed and infected populations. We also have compared all the scenarios to know which is the most effective (see Table 5). Our finding suggests that within the best scenarios analysis, scenario 4 is the most effective, which reduce the massive number of measles incidence and mortality in Bangladesh. However, depending on the availability of fund, other scenario in Table 5 can be considered.

## 4. Optimal control strategy and cost-effective analysis

In this section, we performed optimal control strategy and implemented three time dependent control variables to explore their effectiveness and cost-effective analysis in controlling the spread of measles in the population of Bangladesh. The time dependent control variables $u_1(t)$, $u_2(2)$ and $u_3(t)$ are defined as follows:

i. $u_1(t)$ denotes the distancing control strategy that is the effort at inhibiting the virus transmission from Exposed and infected population. This can be reached through public health advocacy for social distancing, good personal hygiene, diagnosis campaigns, and education programs for public health. Noting that $u_1(t) = 1$ indicates the policy effectively protects against infection, while $u_1(t) = 0$ denotes strategy failure.

**Table 5. Selecting best scenarios in our proposed model of measles control in Bangladesh, for the period 2020–2035.**

| Parameters | Parameter values | Estimated measles actual incident cases | Reduction from baseline | Estimated measles actual mortality | Reduction from baseline |
|---|---|---|---|---|---|
| $\eta$, $\sigma$, $\kappa$ and $\tau$ | Baseline (94%, 93%, 8% & 14%) | $3.714 \times 10^5$ | $0.000 \times 10^5$ | $5.546 \times 10^4$ | $0.000 \times 10^4$ |
| | 96%, 95%, 25% & 25% | $0.444 \times 10^5$ | $3.270 \times 10^5$ | $0.675 \times 10^4$ | $4.871 \times 10^4$ |
| | 98%, 97%, 50% & 50% | $0.012 \times 10^5$ | $3.702 \times 10^5$ | $0.180 \times 10^3$ | $5.528 \times 10^4$ |
| | 99%, 99%, 75% & 75% | $0.297 \times 10^2$ | $3.7137 \times 10^5$ | $0.393 \times 10^1$ | $5.5456 \times 10^4$ |
| | 100%, 100%, 100% & 100% | $0.065 \times 10^1$ | $3.7139 \times 10^5$ | $0.092 \times 10^0$ | $5.5459 \times 10^4$ |
| $\eta$, $\tau$ and $\kappa$ | Baseline (94%, 14% & 8%) | $3.714 \times 10^5$ | $0.000 \times 10^5$ | $5.546 \times 10^4$ | $0.000 \times 10^4$ |
| | 96%, 25% & 25% | $0.447 \times 10^5$ | $3.267 \times 10^5$ | $0.679 \times 10^4$ | $4.867 \times 10^4$ |
| | 98%, 50% & 50% | $0.013 \times 10^5$ | $3.701 \times 10^5$ | $0.183 \times 10^3$ | $5.527 \times 10^4$ |
| | 99%, 75% & 75% | $0.305 \times 10^2$ | $3.7136 \times 10^5$ | $0.404 \times 10^1$ | $5.5455 \times 10^4$ |
| | 100%, 100% & 100% | $0.068 \times 10^1$ | $3.7139 \times 10^5$ | $0.095 \times 10^0$ | $5.5459 \times 10^4$ |
| $\tau$ and $\kappa$ | Baseline (8% and 14%) | $3.714 \times 10^5$ | $0.000 \times 10^5$ | $5.546 \times 10^4$ | $0.000 \times 10^4$ |
| | 25% and 25% | $0.455 \times 10^5$ | $3.259 \times 10^5$ | $0.689 \times 10^4$ | $4.857 \times 10^4$ |
| | 50% and 50% | $0.013 \times 10^5$ | $3.701 \times 10^5$ | $0.188 \times 10^3$ | $5.527 \times 10^4$ |
| | 75% and 75% | $0.311 \times 10^2$ | $3.713 \times 10^5$ | $0.417 \times 10^1$ | $5.5455 \times 10^4$ |
| | 100% and 100% | $0.070 \times 10^1$ | $3.714 \times 10^5$ | $0.098 \times 10^0$ | $5.5459 \times 10^4$ |
| Treatment rate for exposed population ($\kappa$) | Baseline (8%) | $3.714 \times 10^5$ | $0.000 \times 10^5$ | $5.546 \times 10^4$ | $0.000 \times 10^4$ |
| | 25% | $0.524 \times 10^5$ | $3.190 \times 10^5$ | $0.903 \times 10^4$ | $4.643 \times 10^4$ |
| | 50% | $0.031 \times 10^5$ | $3.683 \times 10^5$ | $0.677 \times 10^3$ | $5.478 \times 10^4$ |
| | 75% | $0.002 \times 10^5$ | $3.712 \times 10^5$ | $0.732 \times 10^2$ | $5.537 \times 10^4$ |
| | 100% | $0.003 \times 10^4$ | $3.7137 \times 10^5$ | $0.138 \times 10^2$ | $5.545 \times 10^4$ |

ii. $u_2(t)$ represents the vaccination control strategy, it is assumed that the number of vaccines available during this time period and they are all administrated and used completely. If $u_2(t) = 1$, then the control strategy is effectively used, while $u_2(t) = 0$ means the absence of the control strategy.

iii. $u_3(t)$ indicates control variable to enhance the treatment of infected population with a view to ensure the rapid provision of additional treatment includes providing comfort measures to relieve symptoms and preventing complications. Perceiving $u_3(t) = 1$, then the control strategy is effectively treating the disease, while $u_3(t) = 0$ means the strategy failure.

Subsequently, the optimal control model with the three above-mentioned time-dependent variables is given by the following non-linear differential equations:

$$\frac{dS}{dt} = \mu N - (1 - u_1(t))\beta SI - \eta S - \mu S + \delta I + \rho V_1$$

$$\frac{dV_1}{dt} = \eta S - \rho V_1 - \sigma(1 + u_2(t))V_1 - \mu V_1$$

$$\frac{dV_2}{dt} = \sigma(1 + u_2(t))V_1 - \omega(1 + u_2(t))V_2 - \mu V_2$$

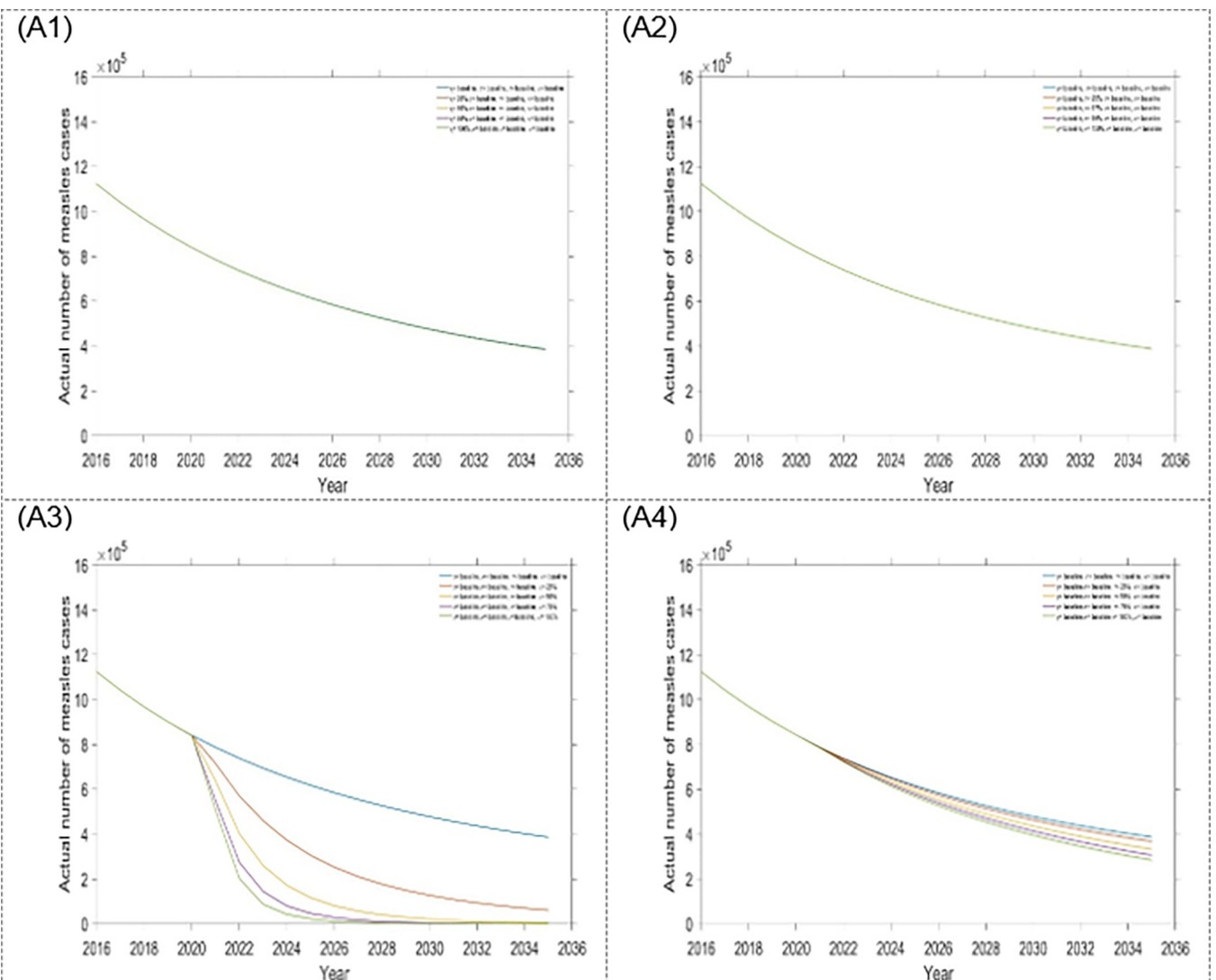

**Fig 4. Impact of the four single intervention strategies on annual incidence of measles: (A1) varying first dose vaccination, (A2) varying second dose vaccination, (A3) varying treatment rate for exposed population, and (A4) varying treatment rate for infected population.**

$$\frac{dE}{dt} = (1 - u_1(t))\beta SI - \alpha E - \kappa(1 + u_3(t))E - \mu E \tag{9}$$

$$\frac{dI}{dt} = \alpha E - \gamma I - \delta I - \tau I - \mu I$$

$$\frac{dR}{dt} = \gamma I + \omega(1 + u_2(t))V_2 + \kappa(1 + u_3(t))E + \tau I - \mu R$$

The goal of presenting the three control variables is to seek the optimal solution required to minimise the numbers of Exposure and infected individuals at minimum cost. Hence, the

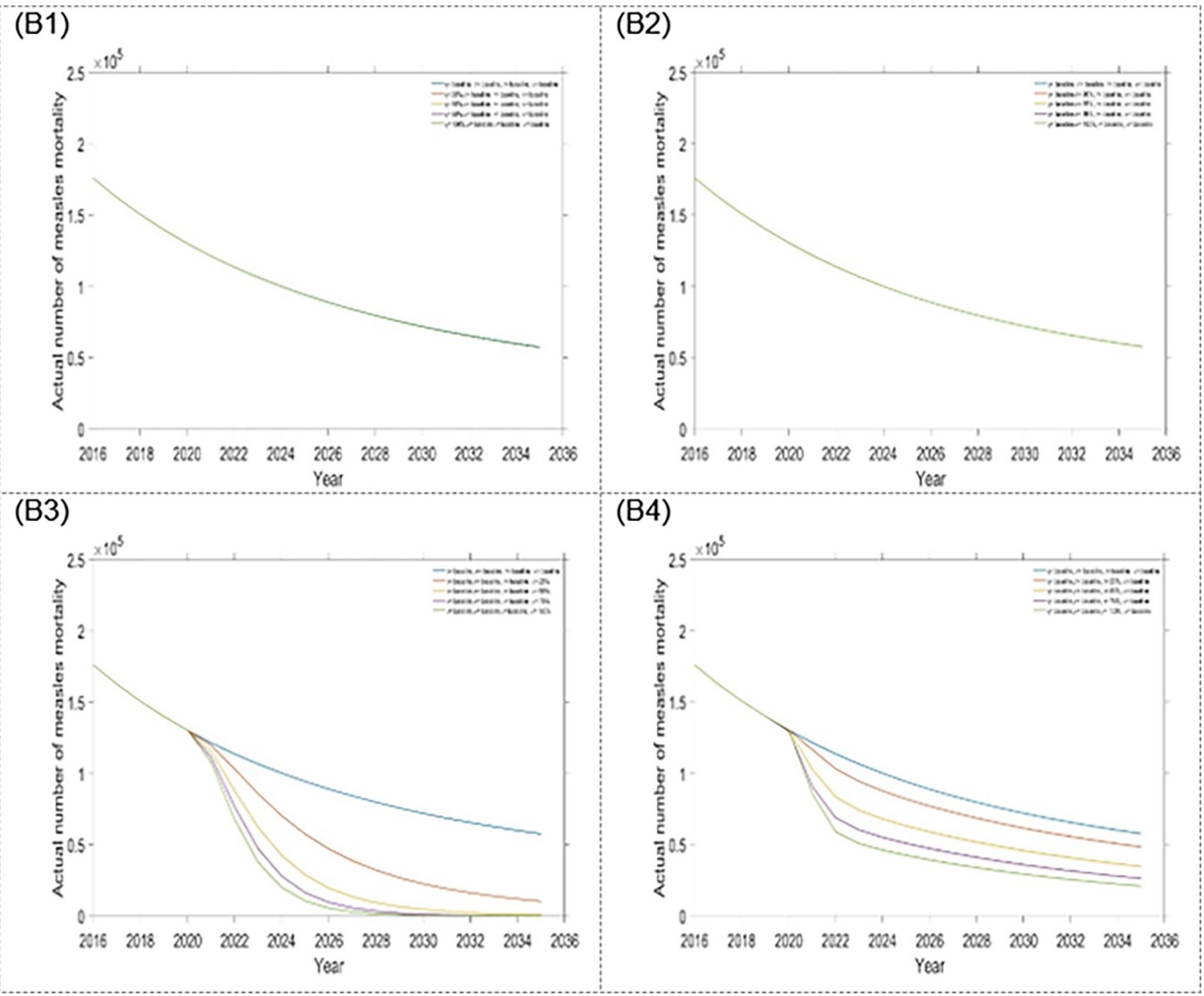

**Fig 5. Impact of the four single intervention strategies on annual mortality of measles:** (B1) varying first dose vaccination, (B2) varying second dose vaccination, (B3) varying treatment rate for exposed population, and (B4) varying treatment rate for infected population.

objective function for this optimal control problem is given by

$$J\left(u_1^*, u_2^*, u_3^*\right) = \min_{0 \le u_1, u_2, u_3 \le 1} \int_{T_0}^{T_f} \left(A_1 E + A_2 I + \frac{1}{2}\left(B_1 u_1^2(t) + B_2 u_2^2(t) + B_3 u_3^2(t)\right)\right) dt, \qquad (10)$$

where, constants $A_i$, i = 1,2 are positive weights essential to balance the objective function. Following other works on infectious diseases control problem [27–30], quadratic cost on the controls are chosen to ensure the control has only one extremum (i.e. maximum or minimum), where $\frac{1}{2} B_1 u_1^2(t)$ is the total cost of executing the distancing, and $\frac{1}{2} B_2 u_2^2(t)$ is the total cost of vaccination and $\frac{1}{2} B_3 u_3^2(t)$ is the total cost of treatment for infected individuals over the time interval $[T_0, T_f]$ (where initial time $T_0 = 0$, final time $T_f = 15$ years period).

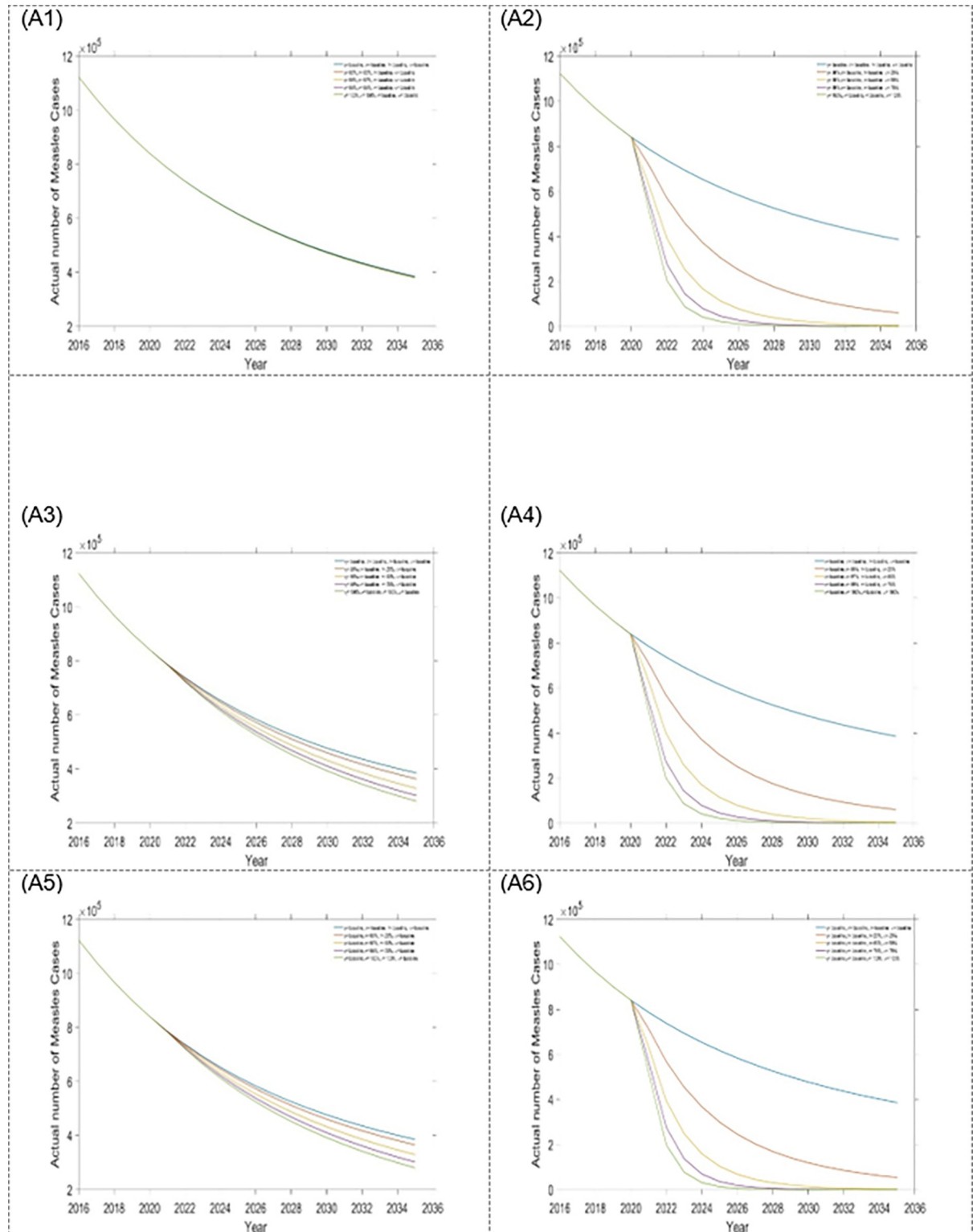

**Fig 6. Impact of the six-double intervention strategies on annual incidence of measles: (A1) η and σ, (A2) η and κ, (A3) η and τ, (A4) σ and κ, (A5) σ and τ, and (A6) κ and τ.**

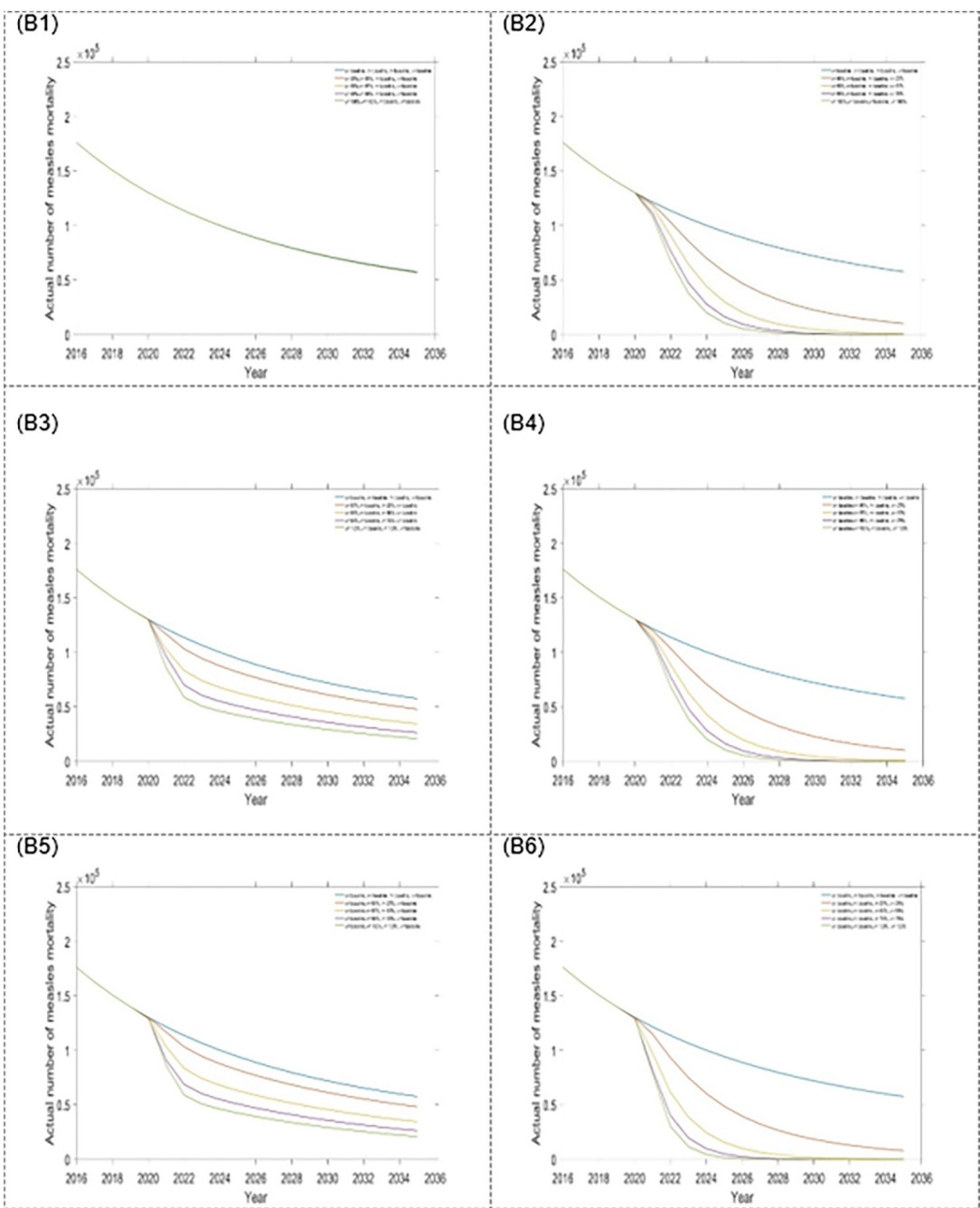

**Fig 7. Impact of the six-double intervention strategies on annual mortality of measles: (B1) η and σ, (B2) η and κ, (B3) η and τ, (B4) σ and κ, (B5) σ and τ, and (B6) κ and τ.**

Precisely, the optimal control strategy $u^* = (u_1^*, u_2^*, u_3^*)$ is required such that

$$J(u_1^*, u_2^*, u_3^*) = \min\{J(u_1, u_2, u_3) : u_1, u_2, u_3 \in U\}, \tag{11}$$

where, U is the non-empty control set defined by

$$U = \{(u_1, u_2, u_3) : (u_1(t), u_2(t), u_3(t)) \text{ are measurable with } 0 \le u_1, u_2, u_3 \le 1 \text{ for } t \in [T_0, T_f]\}.$$

.

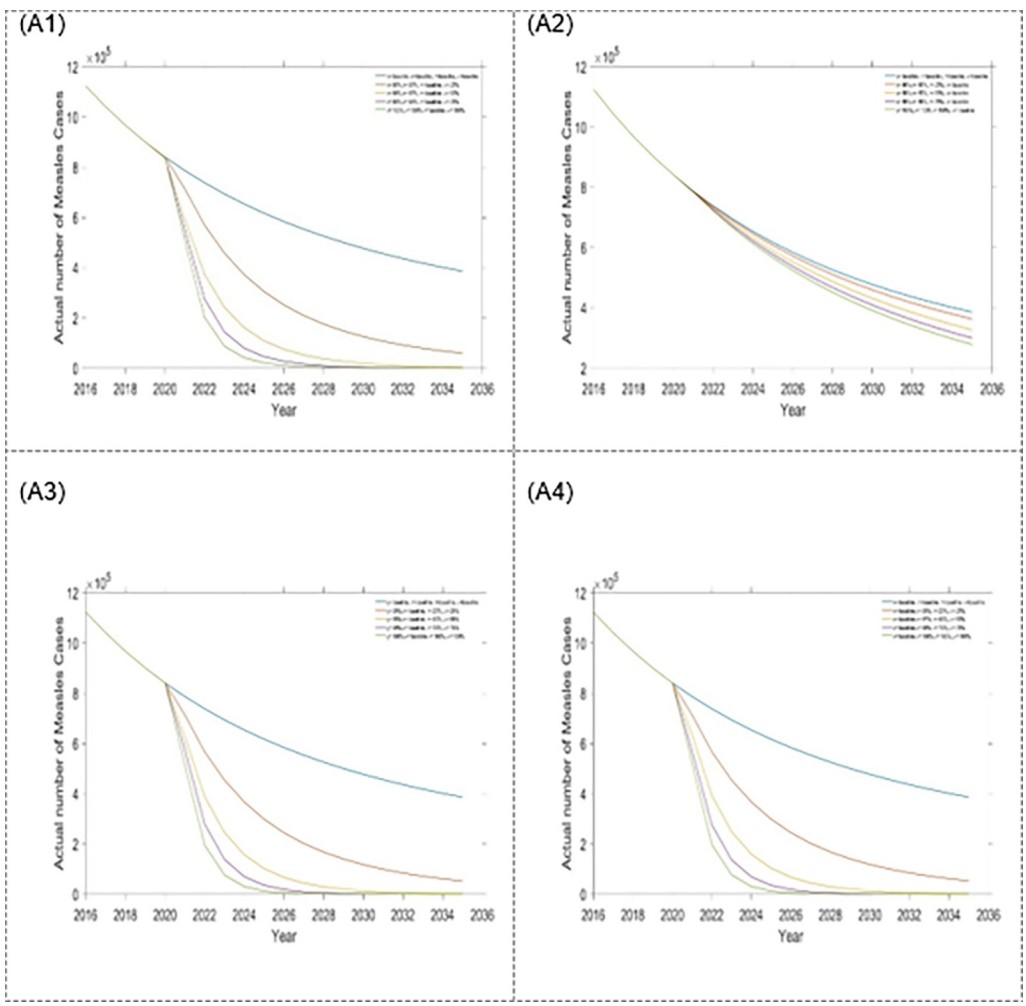

**Fig 8. Impact of the four-triple intervention strategies on annual incidence of measles: (A1) η, σ and κ, (A2) η, σ and τ, (A3) η, τ and κ, and (A4) σ, τ and κ.**

Thus, to regulate the necessary conditions that the optimal control strategy $(u_1^*, u_2^*, u_3^*)$ must satisfy, Pontryagin's maximum principle [31], which changes into the control problem (11) subject to the model (9) that minimising pointwise a Hamiltonian $H_1$, with respect to the control measures. This Hamiltonian is given as

$$H_1 = A_1E + A_2I + \frac{1}{2}(B_1u_1^2(t) + B_2u_2^2(t) + B_3u_3^2(t))$$

$$+\lambda_1(\mu N - (1 - u_1(t))\beta SI - \eta S - \mu S + \delta I + \rho V_1)$$

$$+\lambda_2(\eta S - \rho V_1 - \sigma(1 + u_2(t))V_1 - \mu V_1)$$

$$+\lambda_3(\sigma(1 + u_2(t))V_1 - \omega(1 + u_2(t))V_2 - \mu V_2)$$

$$+\lambda_4((1 - u_1(t))\beta SI - \alpha E - \kappa(1 + u_3(t))E - \mu E)$$

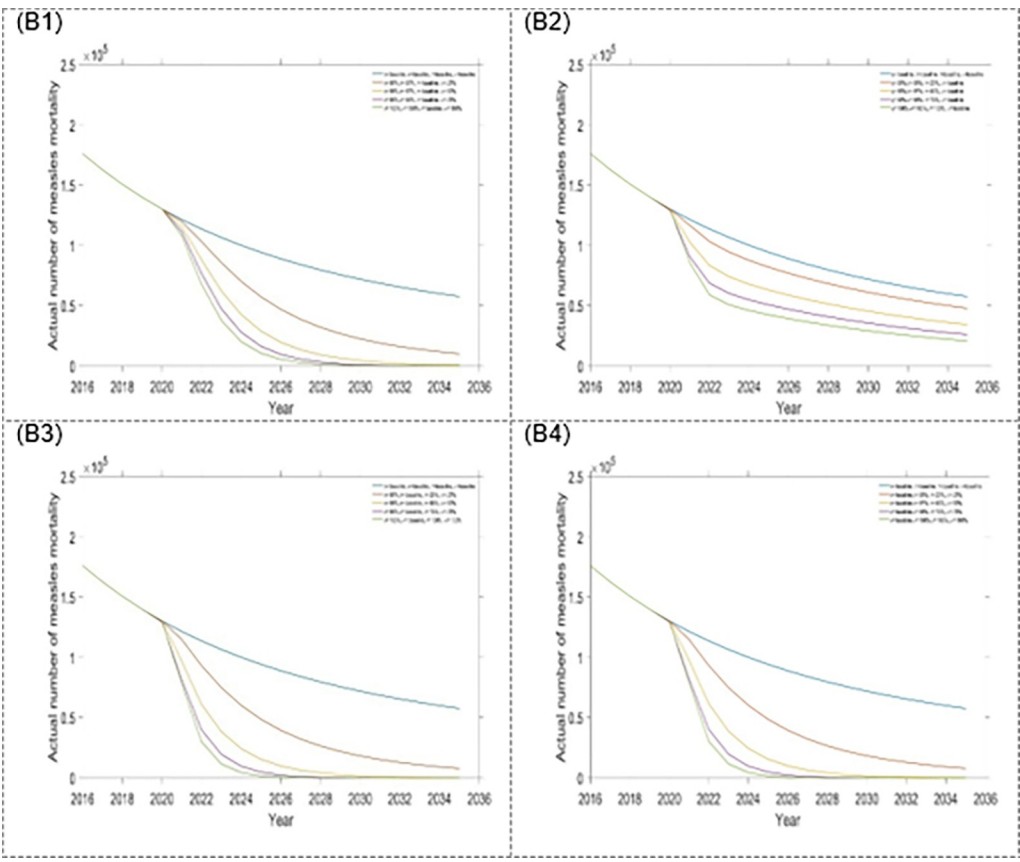

**Fig 9. Impact of the four-triple intervention strategies on annual mortality of measles: (B1) η, σ and κ, (B2) η, σ and τ, (B3) η, τ and κ, and (B4) σ, τ and κ.**

$$+\lambda_5(\alpha E - \gamma I - \delta I - \tau I - \mu I)$$

$$+\lambda_6(\gamma I + \omega(1 + u_2(t))V_2 + \kappa(1 + u_3(t))E + \tau I - \mu R),\tag{12}$$

where, $\lambda_i$, i = 1,2,3,. . .,6, represent the adjoint variables associated with the state variables of

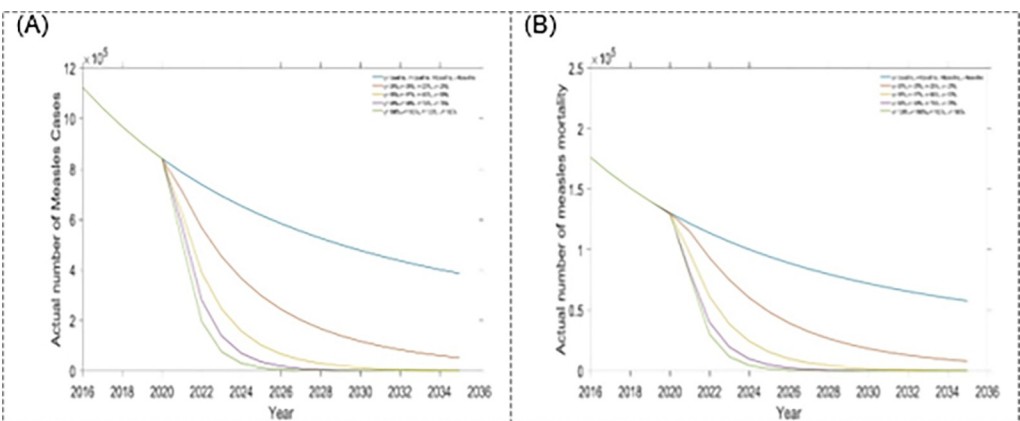

**Fig 10.** Impact of the quadrupled intervention strategies on annual (A) incidence and (B) mortality of measles.

the model (9). The expected outcome for minimising control problem as performed in [30,32] is adapted below. Now using Pontryagin's maximum principle, we obtain the following theorem.

**Theorem:** Given that $(u_1^*, \ u_2^*, \ u_3^*)$ minimises the objective function (10) subject to the corresponding system (9), then the adjoint variables $\lambda_i$, i = 1,2,3,...,6, satisfy the following system

$$\frac{d\lambda_1}{dt} = (\lambda_1 - \lambda_4)(1 - u_1)\beta I + (\lambda_1 - \lambda_2)\eta + \lambda_1\mu$$

$$\frac{d\lambda_2}{dt} = (\lambda_2 - \lambda_1)\rho + (\lambda_2 - \lambda_3)\sigma(1 + u_2) + \lambda_2\mu$$

$$\frac{d\lambda_3}{dt} = (\lambda_3 - \lambda_6)\omega(1 + u_2) + \lambda_3\mu$$

$$\frac{d\lambda_4}{dt} = -A_1 + (\lambda_4 - \lambda_5)\alpha + (\lambda_4 - \lambda_6)\kappa(1 + u_3) + \lambda_4\mu$$

$$\frac{d\lambda_5}{dt} = -A_2 + (\lambda_1 - \lambda_4)\beta S + (\lambda_5 - \lambda_1)\delta + (\lambda_5 - \lambda_6)(\gamma + \tau(1 + u_3)) + \lambda_5\mu$$

$$\frac{d\lambda_6}{dt} = \lambda_6\mu \tag{13}$$

with the terminal (transversality) conditions

$$\lambda_i(T_f) = 0, i = 1, 2, 3, \ldots, 6. \tag{14}$$

Further, the optimal control pair $(u_1^*, \ u_2^*, \ u_3^*)$ is given as follows

$$u_1^* = \max\left\{0, \min\left\{1, \frac{\beta SI(\lambda_4 - \lambda_1)}{B_1}\right\}\right\},$$

$$u_2^* = \max\left\{0, \min\left\{1, \frac{(\lambda_2 - \lambda_3)\kappa E + (\lambda_3 - \lambda_6)\tau I}{B_2}\right\}\right\}$$

$$u_3^* = \max\left\{0, \min\left\{1, \frac{(\lambda_5 - \lambda_6)\tau I}{B_3}\right\}\right\} \tag{15}$$

**Proof:** The existence of the optimal controls $u_1^*$, $u_2^*$ and $u_3^*$ such that

$J(u_1^*(t), u_2^*(t), u_3^*(t)) =_U^{\min} J(u_1, u_2, u_3)$ with state system (9) is given by the convexity of the objective function integrand. By Pontryagin's Maximum Principle [31], the adjoint equations and transversality conditions are obtained. Differentiation of Hamiltonian $H_1$ for the state variables gives the following system,

$$\frac{d\lambda_1}{dt} = -\frac{\partial H_1}{\partial S},$$

$$\frac{d\lambda_2}{dt} = -\frac{\partial H_1}{\partial V_1},$$

$$\frac{d\lambda_3}{dt} = -\frac{\partial H_1}{\partial V_2},$$

$$\frac{d\lambda_4}{dt} = -\frac{\partial H_1}{\partial E},$$

$$\frac{d\lambda_5}{dt} = -\frac{\partial H_1}{\partial I},$$

$$\frac{d\lambda_6}{dt} = -\frac{\partial H_1}{\partial R},$$

with $\lambda_i = 0$, for i = 1,2,3,...,6.

Optimal controls $u_1^*(t)$, $u_2^*(t)$ and $u_3^*(t)$ are derived by the following optimality conditions,

$$\frac{\partial H_1}{\partial u_1} = B_1 u_1 + \lambda_1 \beta SI - \lambda_4 \beta SI = 0,$$

$$\frac{\partial H_1}{\partial u_2} = B_2 u_2 - \lambda_2 \sigma V_1 + \lambda_3 \sigma V_1 - \lambda_3 \omega V_2 + \lambda_6 \omega V_2 = 0,$$

$$\frac{\partial H_1}{\partial u_3} = B_3 u_3 - \lambda_5 \tau I + \lambda_6 \tau I = 0$$

at $u_1^*(t)$, $u_2^*(t)$ and $u_3^*(t)$ on the set U. On this set

$$u_1^*(t) = \frac{\beta SI(\lambda_4 - \lambda_1)}{B_1},$$

$$u_2^*(t) = \frac{(\lambda_2 - \lambda_3)\kappa E + (\lambda_5 - \lambda_6)\tau I}{B_2}.$$

$$u_3^*(t) = \frac{(\lambda_5 - \lambda_6)\tau I}{B_3}$$

This ends of the proof.

Here, we implemented Runge-Kutta fourth-order forward and backward method using MATLAB programming language to solve the subsequent optimality system which consists of (9) and (13) with the characterization (15) within the period of [0, 15] years. The weight constants adopted for balancing the objective function (10) are selected to ensure that no term dominates the other. Therefore, we used equal weight constant for minimising the infectious classes, so that $A_1 = A_2 = 1$. Under other conditions, the weight constants for determining efforts or cost essential to implement the controls are comparatively different, and outcomes in values for $B_1 = 50$, $B_2 = 100$ and $B_3 = 150$ are consistent with previous modelling research. [33]. Details of the numerical procedure for simulating the obtained optimality system are contained [34].

Figs 11 and 12 establish how distancing control, $u_1(t)$ and vaccination control strategies, $u_2(t)$ affect the spread of the measles in Bangladesh. As shown in Figs 11 and 12, to minimise the objective function (10), the optimal control $u_1(t)$ and $u_2(t)$ are continued at the maximum

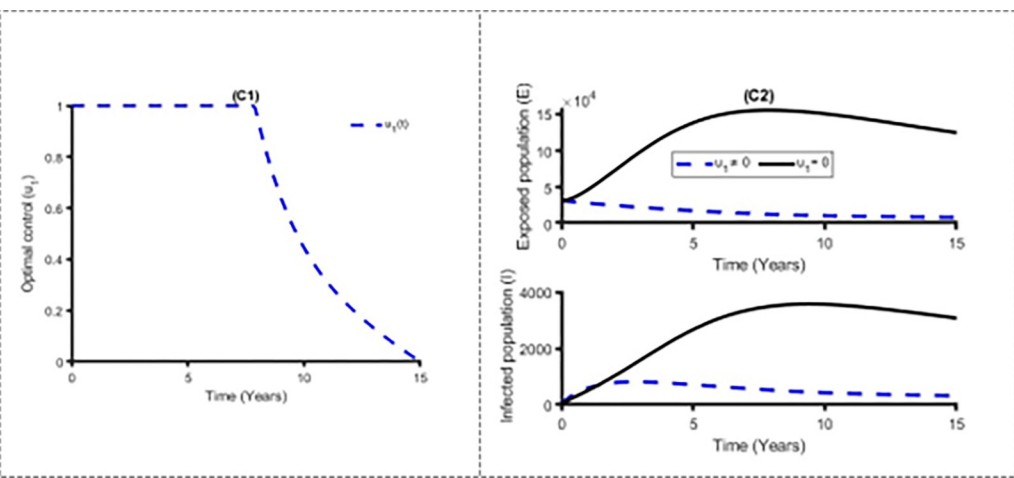

**Fig 11. Control profile $u_1(t)$ and its effects on the measles cases in Bangladesh.**

level (i.e. 100%) for about 8 years and 5 years respectively for Bangladesh population before relaxing to the minimum in the final time. Also as expected, the number of measles infectious individuals are reduced when control is in place. On the other hand, Fig 13 displays the effects of treatment control strategy $u_3(t)$ on the dynamics of measles infection in Bangladesh. We observed that treatment control strategy has small impact on exposed and infected population while distancing control strategy has high impact for reducing the burden of measles cases.

Fig 14 shows the implication of combining the three optimal controls in bringing down the total number of infectious human to zero in Bangladesh. It is observed that optimal solution has achieved when distancing control strategy ($u_1$) is strictly followed to at the maximum level of 100% for 7 years, while the vaccination and treatment control strategies ($u_2$, $u_3$) are at a maximum level above 60% and 20% respectively. It can be seen that the combination of the three control strategies is significantly more effective to decrease the spread of the measles compare to implement each control strategy individually, which is consistent with the previous modelling studies [30,35,36].

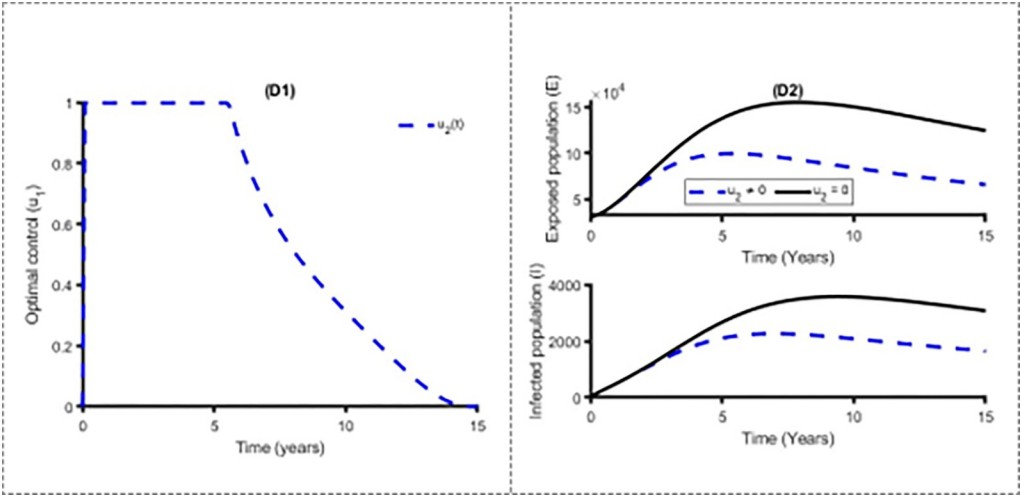

**Fig 12. Control profile $u_2(t)$ and its effects on the measles cases in Bangladesh.**

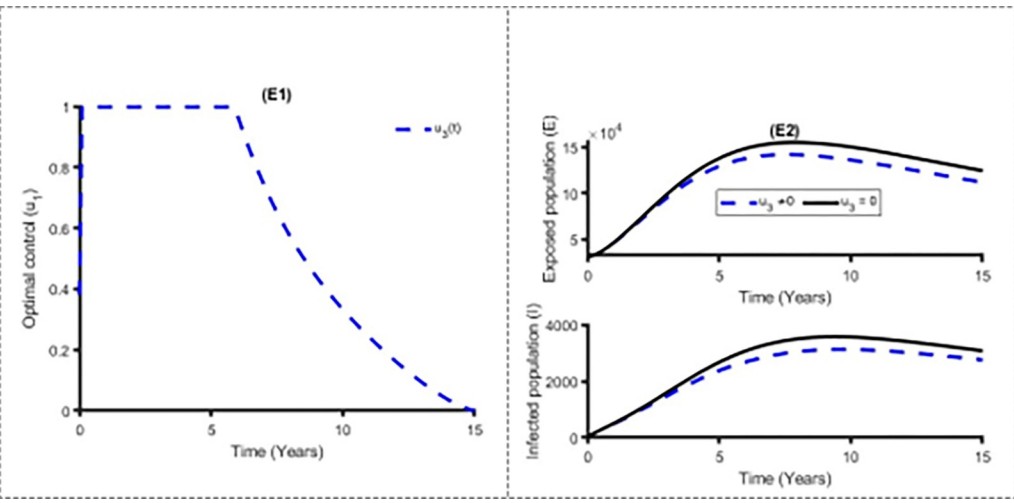

**Fig 13. Control profile $u_3$(t) and its effects on the measles cases in Bangladesh.**

It is crucial to identify the most cost-effective strategy among distancing, vaccination and treatment control as well as their combination control strategies to optimally mitigate the spread of measles at the possible minimum cost. This is performed by associating the differences among each intervention's costs and outcomes; obtained by estimating the incremental cost-effective ratio (ICER), which is defined as the extra cost per additional intervention outcome. Incrementally, when analysing two or more competing intervention policies, one intervention is associated with the next less effective option. The ICER numerator is given by the total difference in intervention costs, active measles cases averted costs and averted productivity losses if applicable, between each scenario and baseline. The ICER denominator is the difference in total number of active measles cases averted. Hence, the following formula obtains the ICER:

$$\text{ICER} = \frac{\text{Difference in total costs between control strategies}}{\text{Difference in total number of active cases averted by control strategies}}, \quad (16)$$

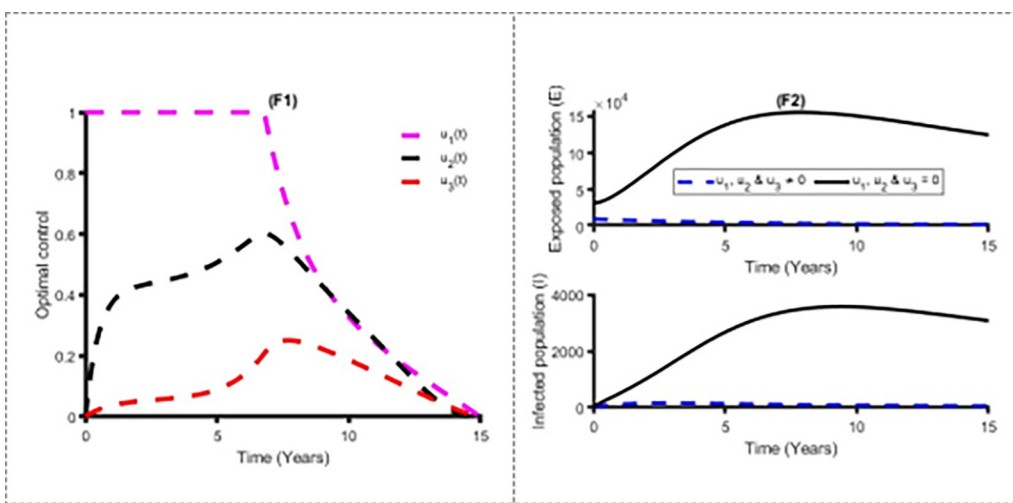

**Fig 14. Control profile $u_1$, $u_2$ & $u_3$(t) and its effects on the measles cases in Bangladesh.**

**Table 6. ICER and ACER in the order of measles cases averted by control measures.**

| Control measures | Total infected averted | Total cost | ICER | ACER |
|---|---|---|---|---|
| $P_1$ | $4.6920 \times 10^6$ | $7.0203 \times 10^5$ | 0.1496 | 0.1496 |
| $P_2$ | $6.5486 \times 10^5$ | $1.5916 \times 10^6$ | −0.2203 | 2.4304 |
| $P_3$ | $4.4332 \times 10^5$ | $2.1402 \times 10^6$ | −2.5934 | 4.8276 |
| $P_{123}$ | $3.2905 \times 10^8$ | $6.6474 \times 10^5$ | −0.0045 | 0.0020 |

We also performed the average cost-effectiveness ratio (ACER), which estimates the effectiveness of a particular intervention technique. The ACER is the ratio between the total cost incurred and the total number of active measles cases averted by that strategy. This is calculated by

$$ACER = \frac{\text{Total cost}}{\text{Total active cases averted}} \qquad (17)$$

The total cost for each of distancing, vaccination and treatment implementation and mutual effort of the optimal control strategy is obtainable from the objective function (10). The cases averted is invaded by computing the difference between infectious individuals with and without control strategy. Let, $P_1$, $P_2$, $P_3$ and $P_{123}$ respectively represent single distancing control strategy $u_1(t)$, single vaccination control strategy $u_2(t)$, single treatment control strategy $u_3(t)$ and the combined effort of the three strategies. Table 6 summarises the ICER and ACER for each and the combination of the control variables $u_1(t)$, $u_2(t)$ and $u_3(t)$ in increasing order of the total infection averted. The ICER and ACER results for $P_1$, $P_2$, $P_3$ and $P_{123}$ are calculated using (16) and shown in Table 6 follows.

Comparing $P_1$, $P_2$, $P_3$ and $P_{123}$ in Table 6 and Fig 15, it is seen that combined control strategy $P_{123}$ is most cost-effective which reduce a significant number of measles cases with low

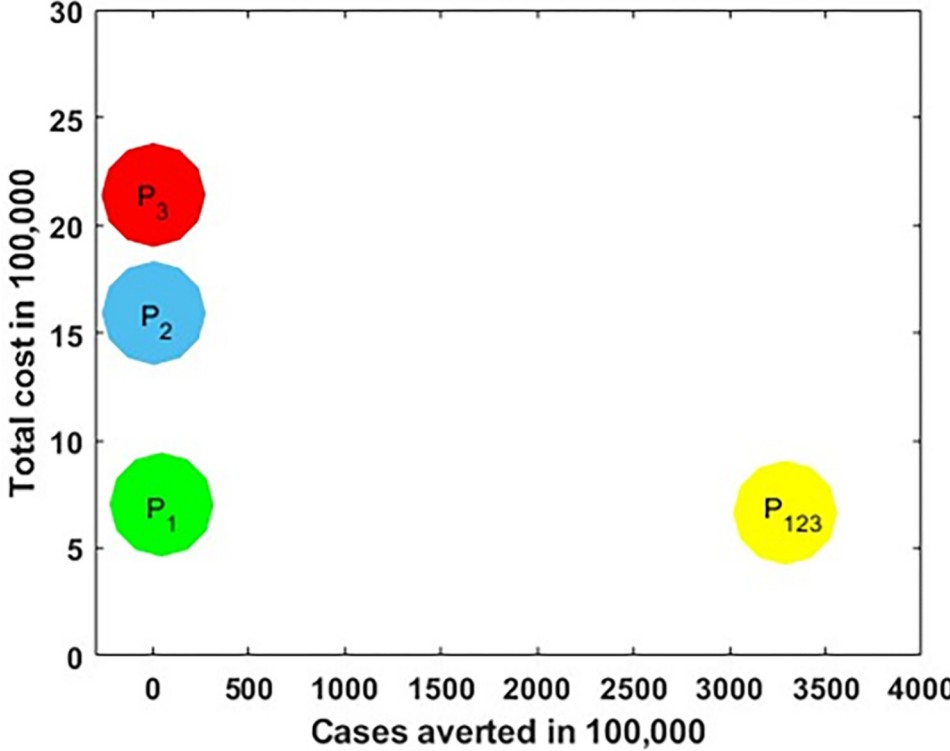

**Fig 15. Comparing cost-effective analysis among $P_1$, $P_2$, $P_3$ and $P_{123}$ control strategies.**

cost compared to $P_1$, $P_2$ and $P_3$ individually, while $P_3$ is the least cost-effective intervention strategy among them.

## 5. Discussion and conclusion

Bangladesh is a resource-poor, high burden measles country compared to other South-East Asian countries, and the transmission dynamics and epidemiology of measles are poorly understood. Therefore, due to paying less attention to the spread of measles, Bangladesh has been suffering from measles for many years even though measles is a vaccine-preventable disease with adequate and timely vaccination [37]. Indeed, ensuring vaccines for all people is a high burden and expensive for Bangladesh, as in other South Asian countries, but still the collected vaccines are not administered properly and therefore a considerable number of vaccines are wasted [38]. Even at the beginning of the last decade, more than half (about 69.7%) of measles vaccines distributed at both ward and village levels almost in all districts were wasted due to improper distribution planning or supply process and lack of proper maintenance [39]. Owing to these problems, the measles vaccination rate was comparatively low than that of other diseases vaccinations [40]. However, subsequently, the Bangladesh government set a target of achieving more than 95% measles vaccination coverage by 2018 to eliminate measles [41].

Finally, following WHO measles elimination strategies, Bangladesh has achieved around 93% coverage with second vaccine doses in every district and delivered through the routine immunization program and financial support from the treatment program and programmatic strength [25]. Although measles control in Bangladesh has significantly progressed–improved first and second doses of vaccine, availability of free treatment services, involvement of multiple partners, newer diagnostic facilities, sufficient human resources, adequate capacity and guidelines–more efforts are required.

In this paper, we presented a measles model with first and second dose vaccine to explore the transmission dynamics of measles in Bangladesh. We derived the basic reproduction number of the measles model and found that basic reproduction numbers play an essential role in the dynamics of measles outbreaks. We fitted measles incidence data with the model from the WHO report to estimate the model parameters. Sensitivity analyses were performed to determine the relative importance of several parameters used in our model. It will help epidemiologists and public health officials focus on the more significant parameters for formulating the measles control strategy. Our analysis led to the observations that transmission rate significantly contributes to the basic reproduction numbers of measles dynamics and needs to be calculated precisely for an accurate outcome.

We acknowledged the importance of comprehensive countrywide programmatic improvements to measles control in Bangladesh. Without such extensive approaches, further rises in the overall disease burden are expected. Four different scenarios were considered (scenario 1, scenario 2, scenario 3 and scenario 4) from combinations of first and second dose vaccines and treatment for exposed and infected populations. Numerous settings were examined to measure the effectiveness of the scenario strategies. In scenario 1, within the four single-intervention strategies, the treatment for the exposed population is the most effective intervention compared to other single-intervention strategies. The result shows that if extensive programmatic development does not occur, especially with the expansion of treatment rate for the exposed population, it is likely that measles will remain uncontrolled in Bangladesh. However, the first and second dose vaccination rates already have covered about 94% and 93%, respectively, in Bangladesh. Therefore, increasing the treatment rate for exposed populations is the better option for controlling measles outbreaks.

Within the six dual-intervention strategies (scenario 2) with treatment for exposed and infected populations is the most effective intervention strategy that reduces the number of incidence and mortality in Bangladesh. Alternatively, activities that aim to improve the first dose of vaccine and exposed or infected population treatment rate are other effective options. However, when we scaled up the treatment rate for the exposed and infected population under scenario 2, the reduction in measles incidence and mortality achieved. Therefore, our outcomes suggest that an improved treatment rate for the exposed and infected population would be insufficient to reduce measles incidence and mortality.

Considering the four-triple intervention strategies (scenario 3), improving the treatment rate for exposed and infected populations with the first dose of vaccination rate is the most effective at measles incidence and mortality reduction compared to other triple intervention strategies. From the analysis of all the scenarios, we found that scenario 4 (first and second dose vaccination rate and treatment rate for exposed and infected population) is the most effective intervention strategy and followed by scenario 3 (first dose vaccination rate and treatment rate for exposed and infected population), scenario 2 (treatment rate for exposed and infected population), and scenario 1 (treatment rate for exposed population). Our findings suggest that the treatment rate for the exposed population is the most critical intervention for decreasing measles incidence and mortality in Bangladesh. However, focus on the first dose and second dose of vaccination rate or treatment rate for the infected population alone will not dramatically affect the decline in measles incidence and mortality in Bangladesh. Taking two or more key interventions simultaneously is the most effective way to reduce measles incidence and mortality in Bangladesh.

Scenarios analysis has been applied in different low and middle income endemic countries to control infectious diseases (e.g. measles, tuberculosis, COVID-19) epidemic [42–45]. Previous studies show that focus on a single intervention strategy will not dramatically affect the decline in infectious disease outbreak but combined two or more interventions is the most effective for reducing the burden of infectious diseases [43,46], which is found to be consistent with our results.Our scenarios describe a variety of potential responses, extending from inaction to extremely ambitious multifactorial strategies. Despite the challenges of delivering effective programmatic measles control in Bangladesh, we believe it is essential to consider such responses. It is because significant impacts from simple public health interventions have previously been demonstrated in resource-limited settings such as Bangladesh. However, the World Health Organization does not presently recommend comprehensive approaches. Our modelling suggests that the high burden of measles incidence and mortality in Bangladesh is likely to increase with the existing programmatic intervention strategy.

We also implemented an optimal control approach via Pontryagin's Maximal principle [31] and formulated the optimal strategies for controlling the measles epidemic in Bangladesh. Three different control strategies were considered including distancing control strategy ($u_1$), vaccination control strategy ($u_2$) and treatment control strategy ($u_3$). Different settings were examined to measure the cost-effectiveness of the control strategies. Between the three-single control strategies, the distancing control strategy ($u_1$) is better in cost-effectiveness than the vaccination and treatment control strategies which reduce a significant number of measles cases in Bangladesh. Therefore, our results suggest that the Bangladesh government should improve distancing control interventions when only one control strategy is used. Naturally, this strategy actively decreases and/or stops the contact between susceptible and infectious individuals of measles. However, combined implementation of distancing, vaccination and treatment strategies is the most cost-effective measure for reducing the burden of measles in Bangladesh.

Optimal control strategies have been applied in other endemic settings to minimise infectious diseases (e.g. measles, tuberculosis, COVID-19) cases and intervention implementation costs. Previous studies show that distancing strategy is the best strategy for the single control strategy implementation to decrease disease burden and intervention costs [33,42,47], which is similar to our results. However, our finding also suggests that combining three control strategy is the most effective way to decrease the measles burden of Bangladesh, consistent with previous works [33,48,49].

In Bangladesh, infectious disease surveillance is not fully recognised and the risk of bias cannot be prohibited. More precise data should be put in place to address alarms related to measles. Precise data leads to better estimation of vital parameters, and this means our projected intervention to decision support is data-dependent. Hence, local and national level policy-makers need to adjust the possibility of under-reporting bias when investigating our results.

## Supporting information

**S1 Data.**
(XLSX)

## Author Contributions

**Conceptualization:** Md Abdul Kuddus, Azizur Rahman, M. Mohiuddin.

**Data curation:** Md Abdul Kuddus.

**Formal analysis:** Md Abdul Kuddus.

**Funding acquisition:** Azizur Rahman.

**Investigation:** Md Abdul Kuddus, Azizur Rahman, Farzana Alam, M. Mohiuddin.

**Methodology:** Md Abdul Kuddus, M. Mohiuddin.

**Project administration:** Azizur Rahman, M. Mohiuddin.

**Resources:** Md Abdul Kuddus, M. Mohiuddin.

**Software:** Md Abdul Kuddus.

**Supervision:** Azizur Rahman.

**Validation:** Md Abdul Kuddus, Azizur Rahman, Farzana Alam, M. Mohiuddin.

**Visualization:** Md Abdul Kuddus, Farzana Alam, M. Mohiuddin.

**Writing – original draft:** Md Abdul Kuddus, M. Mohiuddin.

**Writing – review & editing:** Md Abdul Kuddus, Azizur Rahman, Farzana Alam, M. Mohiuddin.

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
