## [Decision Letter · Decision Letter 0]

4 Nov 2022

PONE-D-21-22479Analysis of the different interventions scenario for programmatic measles control in Bangladesh: a modelling studyPLOS ONE

Dear Dr. Kuddus,

Thank you for submitting your manuscript to PLOS ONE. After careful consideration, we feel that it has merit but does not fully meet PLOS ONE’s publication criteria as it currently stands. Therefore, we invite you to submit a revised version of the manuscript that addresses the points raised during the review process.

Please revise your manuscript in the light of the reviewers' comments. Please be specific and address all comments.

We look forward to receiving your revised manuscript.

Kind regards,

Ejaz Ahmad Khan, M.D, MPH, FFPH

Academic Editor

PLOS ONE

Journal Requirements:

2. PLOS requires an ORCID iD for the corresponding author in Editorial Manager on papers submitted after December 6th, 2016. Please ensure that you have an ORCID iD and that it is validated in Editorial Manager. To do this, go to ‘Update my Information’ (in the upper left-hand corner of the main menu), and click on the Fetch/Validate link next to the ORCID field. This will take you to the ORCID site and allow you to create a new iD or authenticate a pre-existing iD in Editorial Manager. Please see the following video for instructions on linking an ORCID iD to your Editorial Manager account: https://www.youtube.com/watch?v=_xcclfuvtxQ.

4**. **Please include a separate caption for each figure in your manuscript

Reviewers' comments:

Reviewer's Responses to Questions

**Comments to the Author**

1. Is the manuscript technically sound, and do the data support the conclusions?

Reviewer #1: Yes

Reviewer #2: Yes

2. Has the statistical analysis been performed appropriately and rigorously? 

Reviewer #1: Yes

Reviewer #2: Yes

3. Have the authors made all data underlying the findings in their manuscript fully available?

Reviewer #1: Yes

Reviewer #2: Yes

4. Is the manuscript presented in an intelligible fashion and written in standard English?

Reviewer #1: Yes

Reviewer #2: Yes

5. Review Comments to the Author

Reviewer #1: This is a very well written scientific work. It puts together the different arrangements that can either be given individually or in combination. This piece of work would further strengthen if the cost benefit analysis as well as it will be used as a strong advocacy tool.

Reviewer #2: This is an well written article on the topic of measles in Bangladesh. The mathematical models presented in the paper are really interesting. I would leave a minor comment on enriching the discussion with few other similar studies and citing them. You have already discussed your findings in the discussion, it would be better to have some more similar examples from other LMICs.

6. PLOS authors have the option to publish the peer review history of their article (what does this mean?). If published, this will include your full peer review and any attached files.

Reviewer #1: **Yes: **Dr Khalid Nawaz

Reviewer #2: **Yes: **Abdullah Nurus Salam Khan

---

## [Author Response · Author response to Decision Letter 0]

20 Dec 2022

We thank the reviewer for their assessment of our study, and the comments below. We have made changes to the manuscript and replied to other comments below.

Reviewer #1: This is a very well written scientific work. It puts together the different arrangements that can either be given individually or in combination. This piece of work would further strengthen if the cost benefit analysis as well as it will be used as a strong advocacy tool.

Response: Thank you for your valuable comment. We have now added cost benefit analysis of different intervention strategies including distancing, vaccination and treatment to the revised manuscript. Please see page 17-23, line 368 – 521 and page 26-27, line 610-637.

Reviewer #2: This is an well written article on the topic of measles in Bangladesh. The mathematical models presented in the paper are really interesting. I would leave a minor comment on enriching the discussion with few other similar studies and citing them. You have already discussed your findings in the discussion, it would be better to have some more similar examples from other LMICs.

Response: We thank the reviewer for this wonderful suggestion. We have cited some other similar studies in LMICs in the relevant section as well as discussion section to the revised manuscript. Please see page 22-24, line 481 – 488, 527 – 538 and page 26-27, line 595 – 600, 624 – 630.

---

## [Decision Letter · Decision Letter 1]

2 Mar 2023

Analysis of the different interventions scenario for programmatic measles control in Bangladesh: a modelling study

PONE-D-21-22479R1

Dear Dr. Kuddus,

We’re pleased to inform you that your manuscript has been judged scientifically suitable for publication and will be formally accepted for publication once it meets all outstanding technical requirements.

Kind regards,

Jan Rychtář

Academic Editor

PLOS ONE

Additional Editor Comments (optional):

Reviewers' comments:

Reviewer's Responses to Questions

**Comments to the Author**

1. If the authors have adequately addressed your comments raised in a previous round of review and you feel that this manuscript is now acceptable for publication, you may indicate that here to bypass the “Comments to the Author” section, enter your conflict of interest statement in the “Confidential to Editor” section, and submit your "Accept" recommendation.

Reviewer #2: All comments have been addressed

Reviewer #3: All comments have been addressed

2. Is the manuscript technically sound, and do the data support the conclusions?

Reviewer #2: Yes

Reviewer #3: Yes

3. Has the statistical analysis been performed appropriately and rigorously? 

Reviewer #2: Yes

Reviewer #3: Yes

4. Have the authors made all data underlying the findings in their manuscript fully available?

Reviewer #2: Yes

Reviewer #3: Yes

5. Is the manuscript presented in an intelligible fashion and written in standard English?

Reviewer #2: Yes

Reviewer #3: Yes

6. Review Comments to the Author

Reviewer #2: (No Response)

Reviewer #3: Mathematical modeling has played an important role to support planning and evaluation for the larger scale programs. Since the time of measles control in the early 1990s till now, modeling has proved to be an effective tool in the prediction of when threshold breakpoint for elimination has been achieved. The study by Kuddus et al is very important and will make contributions to the current measles infection elimination efforts. The comments related to the above manuscript are indicated below:

Abstract

The abstract is well written and captured all the contents of the manuscript and well structured.

Introduction

I must commend the authors of this manuscript. The introduction section is very well-written.

Materials and methods

- The author has provided in detail the type of modeling functions they have used; however, they have not provided the following.

- I suggest the authors should provide local stability of disease-free and endemic equilibria if possible

- Authors should add more current citations to the one in line 212 e.g., (i) Obabiyi, Olawale Sunday., Akindele Akano Onifade. Global Stability Analysis for Lassa Fever Transmission Dynamics with Optimal Control Application. International Journal of Applied Mathematics, 2018, 31(3), 457 – 482). (ii) Van den Driessche, P., Watmough, J., 2002. Reproduction numbers and sub-threshold endemic equilibria for 438 compartmental models of disease transmission. Math. Biosci. 180: 29-48.

- The legend in Figures 5, 6,7,8,9 and 10 are not clear. Authors should provide clear legend for all Figures

- Author should provide me the code used to fit their model to data as well as code used for all the simulations to ensure that what they present was actual established.

Discussion

The discussion of this manuscript is very well-written with appropriate references cited.

General comments

This manuscript is quite useful in the era of measles elimination. The authors are experts in modeling and this expertise can be appropriately used to support measles elimination effort. I suggest the manuscript should be accepted for publication subject to the aforementioned comments.

7. PLOS authors have the option to publish the peer review history of their article (what does this mean?). If published, this will include your full peer review and any attached files.

Reviewer #2: **Yes: **Abdullah Nurus Salam Khan

Reviewer #3: **Yes: **Akindele Akano Onifade

---

## [Editor Report · Acceptance letter]

19 Jun 2023

PONE-D-21-22479R1 

Analysis of the different interventions scenario for programmatic measles control in Bangladesh: a modelling study 

Dear Dr. Kuddus:

I'm pleased to inform you that your manuscript has been deemed suitable for publication in PLOS ONE. Congratulations! Your manuscript is now with our production department. 

Kind regards, 

on behalf of

Dr. Jan Rychtář 

Academic Editor

PLOS ONE